# Single cell dual-omic atlas of the human developing retina

Zhen Zuo [1,2,8], Xuesen Cheng [1,8], Salma Ferdous[1,8], Jianming Shao[1], Jin Li[1], Yourong Bao[1], Jean Li[1], Jiaxiong Lu[1], Antonio Jacobo Lopez[3], Juliette Wohlschlegel [4], Aric Prieve[4], Mervyn G. Thomas[5], Thomas A. Reh[4], Yumei Li [1], Ala Moshiri[3] & Rui Chen [1,2,6,7] ✉

The development of the retina is under tight temporal and spatial control. To gain insights into the molecular basis of this process, we generate a single-nuclei dual-omic atlas of the human developing retina with approximately 220,000 nuclei from 14 human embryos and fetuses aged between 8 and 23-weeks post-conception with matched macular and peripheral tissues. This atlas captures all major cell classes in the retina, along with a large proportion of progenitors and cell-type-specific precursors. Cell trajectory analysis reveals a transition from continuous progression in early progenitors to a hierarchical development during the later stages of cell type specification. Both known and unrecorded candidate transcription factors, along with gene regulatory networks that drive the transitions of various cell fates, are identified. Comparisons between the macular and peripheral retinae indicate a largely consistent yet distinct developmental pattern. This atlas offers unparalleled resolution into the transcriptional and chromatin accessibility landscapes during development, providing an invaluable resource for deeper insights into retinal development and associated diseases.

As part of the central nervous system, the human retina is a well-organized, multilayered neuronal structure containing seven major cell classes: photoreceptors (cones and rods), amacrine cells (ACs), bipolar cells (BCs), horizontal cells (HCs), retinal ganglion cells (RGCs), and Müller glial cells (MGs)[1]. Furthermore, these cell classes can be divided into approximately 110 distinct cell types[2].

Guided by a multitude of cell fate determinants, these diverse sets of retinal neurons are derived from a common pool of retinal progenitor cells (RPCs)[3] in both sequential and overlapping patterns[4–8]. First, RPCs divide symmetrically to increase the cell population. They then either divide asymmetrically to produce two daughter cells—one

that differentiates and the other that remains a progenitor—or they divide symmetrically into two differentiated daughter cells in later stages[7,9,10]. RPCs can be divided into two major subtypes: primary RPCs (PRPCs) and neurogenic RPCs (NRPCs). PRPCs are typically enriched in cell-cycle associated transcripts, whereas NRPCs are characterized by proneural transcription factors (TFs) that will allow at least one daughter cell to exit the cell cycle and differentiate into a mature retinal neuron[11,12].

RPCs have heterogeneous transcriptomes and competencies before they are specified and differentiated into a particular cell fate[13,14]. This dynamic yet organized differentiation process generally

[1]HGSC, Department of Molecular and Human Genetics, Baylor College of Medicine, 1 Baylor Plaza, Houston, TX, USA. [2]Graduate Program in Quantitative and Computational Biosciences, Baylor College of Medicine, 1 Baylor Plaza, Houston, TX, USA. [3]Department of Ophthalmology & Vision Science, UC Davis School of Medicine, 4860 Y St, Sacramento, CA, USA. [4]Department of Biological Structure, University of Washington, 1410 NE Campus Pkwy, Seattle, WA, USA. [5]Ulverscroft Eye Unit, School of Psychology and Vision Sciences, The University of Leicester, Leicester, UK. [6]Verna and Marrs McLean Department of Biochemistry and Molecular Biology, Baylor College of Medicine, 1 Baylor Plaza, Houston, TX, USA. [7]Gavin Herbert Eye Institute - Center for Translational Vision Research, Department of Ophthalmology, University of California Irvine School of Medicine, Irvine, USA. [8]These authors contributed equally: Zhen Zuo, Xuesen Cheng, Salma Ferdous. ✉e-mail: rui.chen@uci.edu; ruichen@bcm.edu

happens in two phases: in the first phase, RGCs, ACs, HCs, and cones are produced; and in the second, BCs, rods, and MGs are produced[9,15,16]. Also, by *RLBP1* immunolabeling, the most recent publication showed that MGs develop in the presumptive fovea as early as fetal day 59[17], earlier than previously known.

In addition to temporal control, the development of the retina is also under strong spatial control. The macula lutea, a central, cone-rich region unique to simian primates among mammals, allows for higher visual acuity[18–20]. The macula contains a convexiclivate (funnel-shaped) fovea to minimize light scattering and direct light to the surrounding photoreceptors and interneurons[21]. During retinal development, histological and immunohistochemical investigations demonstrated that the central regions are developmentally accelerated[9]. As early as post-conception week (PCW) 20, the presumptive fovea can be identified as a single layer of cones and the foveal pit begins to appear at roughly PCW 25[22,23]. The location of the presumptive fovea is characterized by a rod-free region that is likely formed through the inhibition of rod photoreceptor development in that region[24,25]. Although the exact molecular developmental mechanisms underlying foveal formation need further elucidation, a recent study in chick retina suggests the involvement of the retinoic acid (RA) and *FGF8* pathway in the formation of the high-acuity area[26].

In this study, we generated a detailed developmental atlas at single-cell resolution by profiling approximately 220 K single nuclei in both the developing macular and peripheral retinae from fourteen human embryos and fetuses spanning PCW 8 to 23 using dual-omic RNA-seq and ATAC-seq. Integrative analysis of this dataset allows us to identify and map the developmental trajectory of over 60 cell classes, along with candidate TFs and gene regulatory networks (GRNs) driving the development of each major cell class. Additionally, genes and pathways that differ between the macular and peripheral retinae are evaluated in order to provide additional insights into macular formation in humans.

## Results

### Overview of the sn-dual-omic human developing retina atlas

To characterize the dynamics of gene expression and regulation during human retinal development, 10X Chromium single-nuclei dual-omic ATAC-seq and RNA-seq were performed on 28 human tissues at six different time intervals (Fig. 1A and Supplementary Data 1). In total, 315,177 nuclei were profiled, with 226,506 nuclei remaining after quality control (Supplementary Data 2). After annotation, nuclei were classified into nine major classes according to their transcriptomes. The Uniform Manifold Approximation and Projections (UMAPs) of the single-nucleus transcriptome (Fig. 1B) and chromatin accessibility (Fig. 1C) demonstrate distinct clusters amongst different major classes. Cell annotations were subsequently visualized and validated with developmental markers obtained from previous studies[27] (Supplementary Fig. 1H and Supplementary Fig. 3). To assess the consistency between the RNA-seq and ATAC-seq data, gene expression (Fig. 1D) and gene score (Fig. 1E) heatmaps were generated for the top 50 differentially expressed genes of each major class (Supplementary Data 3). Gene scores serve as a quantitative measure of the extent of gene-level chromatin accessibility. As expected, the gene scores aligned well with gene expression, which indicates consistency. By overlaying our data with the adult retina reference atlas, a total of 61 cell types, including 5 precursor groups, were identified, while 7 out of the 60 identified in adults could not be annotated in the developmental data (Supplementary Fig. 2H and Supplementary Data 4). Cell types were then further grouped into subclasses (Fig. 1F).

Based on the annotations, the cell proportion was calculated, and the cell birth rate was estimated (Supplementary Fig. 4). As expected, at the initial time point (PCW 8), the sample exhibits the highest proportion of RPCs, approximately 60% (Supplementary Fig. 4A). It is worth noting that a few MGs and BCs have can be detected at as early

as PCW 8 (Fig. 1G and Supplementary Fig. 1I), which is consistent with previous immunolabeling results[17]. Interestingly, Fig. 1G shows that a variation in the sequence of major class birth is observed between the macula and periphery. Specifically, a significant number of BCs developed before rods in the macula, whereas in the periphery, a significant number of rods appeared before the BCs. As expected, compared with the periphery, a significant number of differentiated neurons emerge earlier in the macula (Fig. 1G). To further quantify the time delay, a maturation score for each major class in the macular and peripheral retinae across time points was calculated (Fig. 1H). Consistently, the maturation scores in both locations increase over time, and for the same time point, the maturation scores are consistently larger in the macula. This maturation score is further supported by the temporal expression pattern of known marker genes[28]. For example, photoreceptor marker expression consistently increases during photoreceptor maturation (Fig. 1I). However, distinct time delays are observed between the peripheral and macular retinae across various cell types. When comparing the major-class-specific abundance, a larger delay between the macula and periphery is observed for late-born major classes, such as BCs and MGs, compared to early-born cell types (Fig. 1J). For instance, the average timing of MGs in the macula is PCW 17, while in the periphery, it is PCW 22. Conversely, the average timing for RGCs is approximately PCW 13 in both the macula and periphery.

### PRPC gene expression revealed unique modules

RPCs serve as the origin for all the various cell types found in the neural retina. Within the progenitor cluster, cells can be divided into PRPCs, NRPCs, and MGs (Supplementary Fig. 1G). RPCs exhibited gene expression patterns that were temporally and spatially regulated. Both PRPCs and NRPCs consistently displayed gene expression and chromatin accessibility heterogeneity from early to late time points in both the macular and peripheral regions (Fig. 2A and Supplementary Fig. 1J). Utilizing velocity estimates derived from gene expression and chromatin accessibility, it was observed that PRPCs exhibit bidirectional flows, with one direction leading towards NRPCs and the other towards MGs (Fig. 2B). Furthermore, to enable the ordering of nuclei from early to late stages, latent time was calculated based on PRPC velocities (Fig. 2C). As anticipated, latent time was significantly positively correlated with age (Pearson correlation 0.759, with an adjusted *p*-value of 0.000, see source data) (Fig. 2D). Additionally, as expected, using two-sample t-tests (see source data), nuclei in the macula consistently demonstrated a significantly larger average latent time compared to their peripheral counterparts for all the time points, except for PCW23, which had too few nuclei in the macula.

After ordering PRPCs from early to late based on latent time, the heterogeneity of genes underlying PRPC development was quantified. Subsequently, 2860 latent-time-correlated genes were identified and among them 2055 can be further grouped into three distinct modules (Fig. 2E and Supplementary Data 5). Genes in Module 1 were predominantly expressed during the early PRPC stage, while genes in Module 2 became prominent during the transition from early to late PRPCs, and genes in Module 3 were notably expressed at later PRPC stages (Fig. 2E). Consequently, genes in Module 1 exhibited decreasing expression, genes from Module 3 steadily increased in expression over time, and genes in Module 2 exhibited a peak in the middle (Fig. 2F). Gene ontology (GO) analysis (Fig. 2G and Supplementary Data 5) revealed that Module 1 genes are enriched in early development-related terms such as cell adhesion, system development, morphogenesis, and axon guidance; Module 2 genes are associated with cytoplasmic translation and peptide biosynthetic processes; and Module 3 genes are linked to neurogenesis and the generation of neurons.

Next, an attempt was made to link the heterogeneity of genes underlying PRPC development with biased fate. During development,

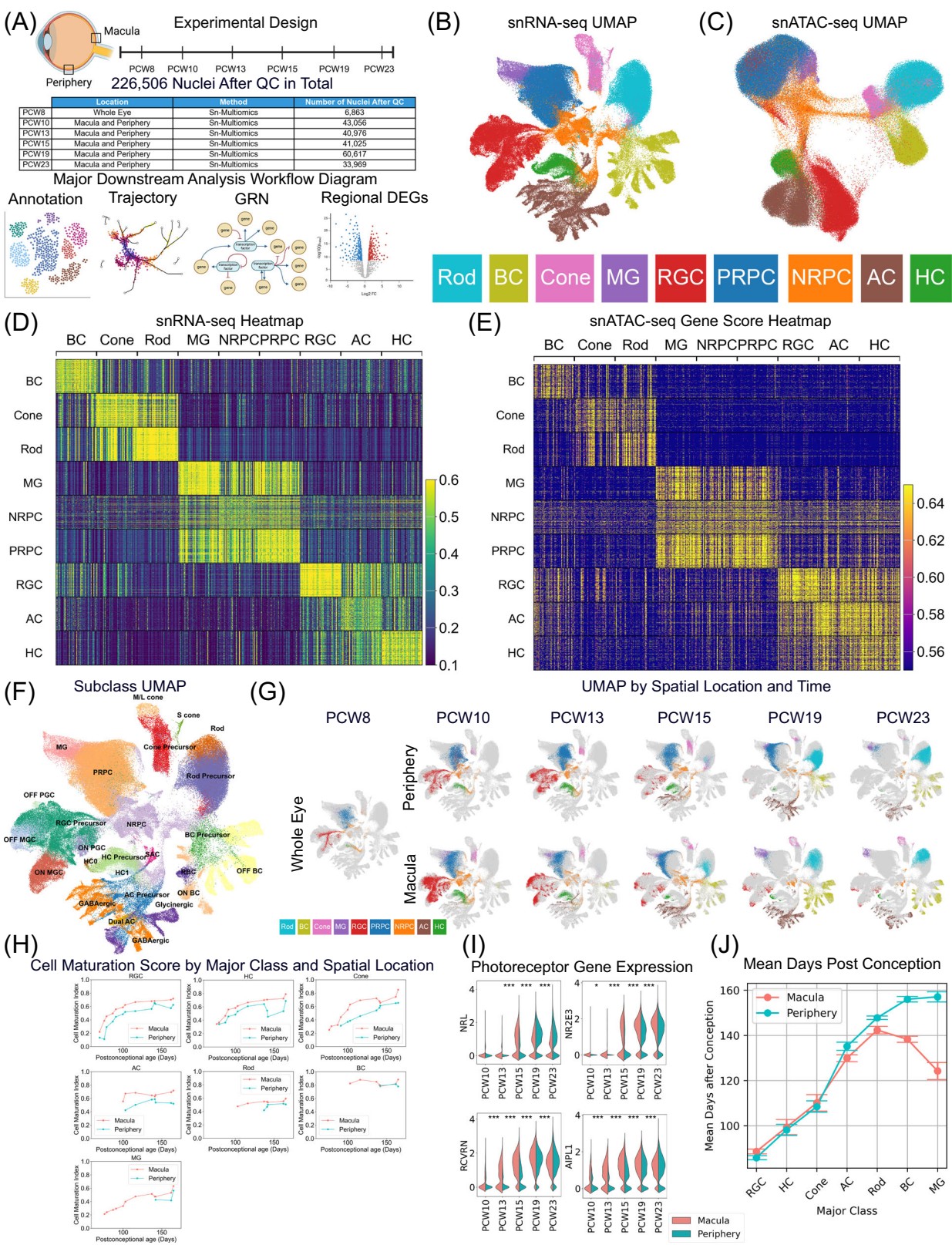

PRPCs can transition into either NRPCs or MGs. Those transitioning cells can be identified based on cell trajectory and fate probabilities. Using velocity analysis, a subset of PRPCs was identified as transitioning to NRPCs (Fig. 2H). These proneural PRPCs overlapped with PRPCs exhibiting a high Module 1 score. More specifically, NRPCs fate probabilities and Module 1 module score show a Pearson Correlation Coefficient of 0.589 and a *p*-value of 0.000 (see source data). This signifies there is an elevated expression level of genes associated with Module 1 in cells transitioning into NRPCs. As expected, higher expression of retinal neurogenesis genes is observed in this transient cell population. One example is *FOXN4* (Fig. 2H), which plays a crucial role in the formation of HCs and ACs in the vertebrate retina[29]. Notably, among cells destined for NRPC transition, a list of genes associated with progenitor maintenance and proliferation were identified. For

**Fig. 1 | Overview of the single-nuclei dual-omic atlas of the human developing retina. A** The study design of this work. Samples were collected from either (1) whole retina at PCW 8, or (2) macula and periphery of the same retina from PCW 10 to PCW 23. Subsequently, with a total of 28 samples from 14 human embryos and fetuses, gene expression and open chromatin profiling from the same nuclei was performed using the 10X Chromium sn-dual-omic ATAC + Gene Expression technology (Supplementary Data 1). The bottom panel shows analysis workflow diagrams. **B** UMAP of RNA-Seq data colored by annotated major classes. **C** UMAP of ATAC-Seq data colored by annotated major classes. **D** Heatmap of the top 50 differentially expressed genes across each annotated major class (Supplementary Data 3). Each row is a nucleus and each column is a marker gene. **E** Heatmap of ATAC gene scores for the differentially expressed genes (same genes and nuclei with the same order as Fig. 1D) grouped by annotated major class. **F** UMAP of RNA-Seq data colored by annotated subclasses. **G** UMAP is separated by location and clock time. Dots were colored by annotated major class, and gray dots are background nuclei that are not from the corresponding time and location pair. **H** Line

graphs of cell maturation scores against sample age (PCW) for each major class, colored by the macula and periphery. The cell maturation score represents the degree of a cell's similarity to cells in the corresponding adult data. **I** Violin plots of previously published photoreceptor-maturation gene expression[28] between macula and periphery. To compare gene expression values temporally, two-sided overestimates variance t-test[59] was applied to different PCW groups among 47,446 photoreceptors. The Benjamini-Hochberg procedure was applied. **J** Line graphs of average cell age for each major class, colored by macula and periphery. Cells were down-sampling equally by PCW groups, and cell age was defined as sample's age. To calculate means and error bars, 41,984 RGCs, 9532 HCs, 9324 cones, 26,384 ACs, 38,167 rods, 20,996 BCs, and 6553 MGs were used. Error bars represent the 95% confidence intervals from two-tailed t-tests. The Benjamini-Hochberg procedure was applied. Source data are provided as a Source Data file. Panel A was created with BioRender.com and released under a Creative Commons Attribution-NonCommercial-NoDerivs 4.0 International license (https://creativecommons.org/licenses/by-nc-nd/4.0/deed.en).

example, *ASPM* has been documented for its specific role in preserving symmetric proliferative divisions of neuroepithelial cells[30]. Moreover, in those NRPC transitional cells, both consistent and inconsistent gene expression patterns were found between humans and mice. For example, *ECT2*, a gene linked to tumor progression, has been shown to be expressed in a subset of mouse PRPCs and used as an example to show mouse RPC heterogeneity[31]. Consistently, *ECT2* also showed heterogeneity in human RPCs and is expressed exclusively in PRPCs likely fated to become NRPCs (Fig. 2H). On the other hand, human-specific genes such as *ARHGAP11B* exhibited higher expression levels in the transitioning PRPCs. *ARHGAP11B* has been reported as a driver for human basal progenitor proliferation[32]; this gene has no homologous gene in mice and is unstudied in human retina development.

Finally, to identify candidate TFs driving PRPC development, GRN analysis based on the transcriptome, open chromatin, and TF-target gene interactions was performed using the Pando[33] software tool (Fig. 2I, Supplementary Data 6). Among them, *MXD3* and *NPAS3*, two genes encoding basic helix-loop-helix proteins, were identified. Both *MXD3*[34] and *NPAS3*[35], have been shown to play a critical role in brain development, but are not well studied in retinal development. Several other TFs identified in our samples, such as *ZNF367*[36], *MECOM*[37], and *PROX1*[38] have been associated with the differentiation of RPCs.

## Distinct gene expressions in varied NRPC fates

After assessing heterogeneity within PRPCs, heterogeneity within NRPCs was examined. As the transition cell state, NRPCs give rise to all retinal neurons. Interestingly, unlike the continuous distribution observed for PRPCs, distinct clusters are observed for NRPCs, suggesting distinct subgroups might exist.

To test if clusters of NRPCs correspond to distinct neuronal fates, velocity analysis (Supplementary Fig. 5A) was used to infer cell fate by estimating cell fate probabilities to various neuron types (Fig. 3A). Cells with a high probability of developing into a neuronal type were assigned as the progenitors of the corresponding type (Fig. 3B). To evaluate the fate inference performance, the identified birth timings were compared to previous knowledge. Indeed, progenitors for RGCs, HCs, ACs, and cones emerged earlier than PCW 13 (Fig. 3C). Later, rod and BC progenitors occurred from PCW 15 to 23. As expected, macular NRPC specification precedes the peripheral ones. For instance, cone progenitors are prevalent in the periphery from PCW 10 to 15; however, in the macula, cone progenitors are barely observed at or after PCW 13. By calculating NRPC birth rate, we found that major classes such as BCs and ACs showed greater developmental delay in the periphery compared with their macular counterparts than other major classes (Fig. 3D). The inference is further supported by examining the expression pattern of TFs known to drive specific cell fate determination (Fig. 3E). For example, upon *OTX2* activation, NRPCs predominantly differentiate into photoreceptors and BCs[39,40], which is

consistent with our results showing *OTX2* expression exclusively in NRPCs that adopt a BC, rod, or cone fate (Fig. 3E). The identification of NRPC clusters specific for each retinal neuron type allows identification of potential genes (Fig. 3F) and pathways driving neuron type specifications (Fig. 3G and Supplementary Data 7). With the GO analysis, in addition to identifying shared biological process terms such as "neuron differentiation" and "neuron development," specific class-related terms were found. Notably, terms like "postsynapse organization" were associated with NRPCs destined for RGCs, while terms related to "visual perception" were prominent in NRPCs committed to rods.

To gain insights of gene regulation during initial cell fate specification, GRNs were constructed based on the NRPC dual-omic profile. In total, 95 TFs were identified which regulate 532 target genes through 9620 unique regions (Supplementary Data 6). TFs were grouped into 2 distinct modules based on their regulatory effects on target genes (Fig. 3H). Among them, module 1 is predominantly enriched with TFs related to the BC/cone/rod fate, of which 11 have been reported. In contrast, TFs in module 2 mainly promote the RGC/AC/HC fate, including 5 RGC/AC/HC related TFs. Based on the TF expression pattern in NRPCs, the predicted specification function of each TF was assigned to a specific major class (Table 1, Supplementary Data 8). In addition, among them, 38 out of 95 have already been demonstrated to play a role in retinal neuron development while the rest (57 TFs) are undocumented TFs (Supplementary Data 8). After removing TFs promoting MG genesis, 22 out of 32 (68.75%) functional predictions match with previous knowledge from loss of function experiments, while 10 TFs were shown to have specification functions in major classes other than the predicted ones. Most of the cell fate incorrect predictions happened for the rod progenitors and TFs enriched in the rod progenitors actually promote cone, BC, and RGC specification.

## Formation of AC subclasses from NRPCs at various times

As described above, progenitors for each distinct retinal neuronal class share a common cluster of NRPCs before committing to a differentiated cell fate. With over 60 cell types identified in our study, we can investigate the developmental trajectory at the subclass level to identify regulation that happens in the later part of differentiation. For example, ACs can be classified into multiple distinct subclasses, including Starburst ACs (SACs), GABAergic ACs, Glycinergic ACs, and dual ACs. By performing clustering and trajectory analysis of the AC branch, a consistent flow from AC progenitors to AC precursors to committed AC subclasses was observed (Fig. 4A). Consistent with previous research[41], a chronological development of SACs, GABAergic ACs, dual ACs, and Glycinergic ACs was observed (Figs. 4B and C).

To identify gene regulation driving AC development, GRN analysis was performed on AC progenitors, AC precursors and mature ACs, resulting in two distinct gene modules (Fig. 4D and Supplementary

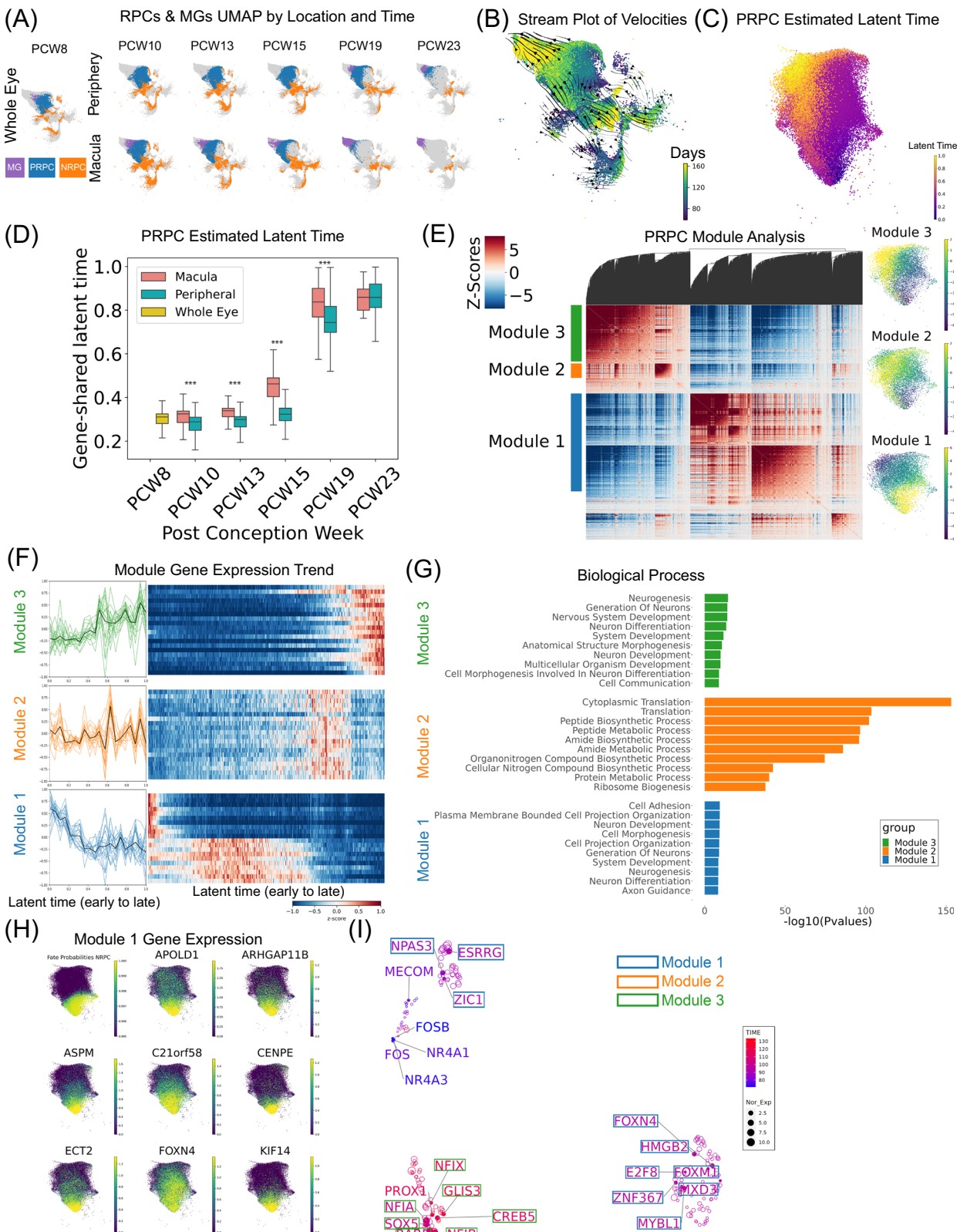

Data 6). TFs in Module 1 that are enriched in AC progenitors, and TFs in Module 2 that are enriched in mature ACs or AC precursors (Supplementary Fig.7A). Consistently, Module 1 includes TFs responsible for specifying AC fate, such as *PTF1A*[42] and *PRDM13*[43], and TFs involved in early AC subclass specification, including *ONECUT1* that determines SACs fate[44]. Additionally, distinct regulatory effects by TFs on target genes were discerned; for instance, *ONECUT1* was predicted to exert a

positive regulation on *PROX1*, *DACH1*, *ZFPM2*, and *HIF1A*, while concurrently displaying a negative regulation on *MEIS2* (Fig. 4E). The same analysis was repeated for all major classes to identify GRNs that drive development (Supplementary Data 6 and Supplementary Fig. 6).

A striking observation is that distinct transcriptomic profiles (Fig. 4F) and chromatin landscapes (Supplementary Fig. 7B) become evident in AC progenitors. Therefore, distinct molecular biases are

**Fig. 2 | PRPC gene expression reveals unique modules with distinct biological processes. A** UMAP of RPCs and MGs separated by spatial location and clock time. **B** UMAP of RPCs and MGs with a stream plot of estimated velocity vectors on top. **C** UMAP of PRPCs colored with estimated latent time. Smaller latent time indicates nuclei that are at an earlier developmental stage, while larger latent time indicates nuclei that are at a later developmental state. **D** Box plot of estimated latent time against PCW for PRPCs. The error bars represent the maximum and minimum values. The box spans the 25th to 75th percentile, with the line inside the box indicating the median. To test if PCW and latent time are correlated, two-sided Pearson's product-moment correlation test was applied (adjusted *p*-value 0.000). For each PCW group, to test if macula and peripheral have the same average latent time, a two-sided Welch two sample t-test was applied. In total, 52,479 cells were used for the tests. The Bonferroni correction was applied for the tests. **E** Correlation heatmap of latent-time-correlated genes (2,860 genes in total and 2,055 can be further grouped into three distinct modules). UMAPs on the right represent module-specific cell module scores. **F** Gene expression trends of 20 genes within each gene module, ordered by latent time. Within each module, genes were sorted by gene module score, which represents the likelihood of a gene belonging to a module. For each module, the smoothed z-scores of gene expressions against latent time were plotted on the left. The black lines in the middle are averaged expression values. **G** Gene Ontology analysis of gene modules. To measure the significance of a functional term, the one-sided hypergeometric test was used with the input gene list (100 genes in each module). Top 10 significant biological processes were plotted with adjusted *p*-values from Benjamini-Hochberg procedure. **H** UMAP of PRPC estimated fate probability for NRPC fate. UMAPs for gene expression of Module 1 genes. **I** PRPC GRN showing key regulators. UMAP was built on similarities of regulation effects on target genes. TFs were labeled by gene module information and colored by gene expression weighted time. The sizes represent normalized gene expressions.

established, showing a biased fate probability to different mature AC subclasses (Supplementary Fig. 7C, 7D, 7I, and 7J), which may lead to distinct groups of mature AC subclasses. To further elucidate the differences between early and late AC NRPCs, we analyzed molecular variations (Supplementary Figs. 7 E-H). In total, 10,910 peaks, 2491 genes with varying accessibility, 3643 genes with different expressions, and 86 motif patterns were identified (Supplementary Data 9). Among those, 45 TFs with consistent gene expression changes and motif enrichment across developmental time were identified (Fig. 4G). Indeed, several TFs known to play an important role in AC development were identified. For example, *ONECUT1* has decreasing expression from early to late AC progenitors (Fig. 4H). Consistent with gene expression, the accessibility of peaks containing the *ONECUT1* motif varied over time and *ONECUT1* binding is most prevalent at PCW 8 (Fig. 4I).

## Velocity models identify different epigenome transcriptome interaction patterns

With the single nuclei dual-omic data, it is possible to systematically assess chromatin accessibility and their association with gene expression. In total, 400,355 open chromatin regions were identified (Supplementary Fig. 8A and Supplementary Data 10). Among them, 70,033 are cell-type-specific differentially accessed regions (DARs) (Fig. 5A and Supplementary Data 10). Interestingly, almost no NRPC-specific DARs are observed, consistent with the rapid transitional nature of NRPCs. In the meanwhile, less than half of DARs are development-specific and not observed in the adult (Fig. 5B). These developmental specific DARs are more closely and specifically associated with developmental processes. For example, developmental-specific RGC DARs are enriched in genes associated with "RGC Axon Guidance" and "RGC cell differentiation" (Table 2).

The correlation between chromatin accessibility and gene expression was calculated to link specific open chromatin regions (OCRs) to their potential targeted genes (Supplementary Fig. 8B). In total, 40,576 peak-gene pairs are identified involving 28,311 OCRs and 7586 genes (Supplementary Data 10). Consistent with the idea that these linked OCRs were likely to be gene regulatory elements, a significantly higher portion of linked OCRs overlap with active enhancer elements based on histone modifications (Fig. 5C). Specifically, 8% of the linked OCRs are positive for both H3K27ac and H3K4me2, compared to 6% for all OCRs. About one-fourth of linked OCRs (6796/28,311) are major-class-specific (Fig. 5D). Notably, we found that PRPCs have the greatest number of linked OCRs (1672) among all major classes, while NRPC have the least. As an example, *OTX2* exhibits high expression in photoreceptors and BCs (Fig. 5E). Consistently, upon comparing the chromatin accessibility among different major classes, we observed that the *OTX2* cis OCRs are more accessible within those major classes (Fig. 5F). Furthermore, five out of twelve orthologous regions of previously reported *OTX2* cis-regulatory modules in mice[45]

are identified as linked peaks in our analysis, along with additional undocumented candidate OCRs.

Based on the relationship between chromatin state and gene expression, multiple phases of gene regulation can be classified, such as priming, coupled-on, coupled-off, and decoupled for each gene[46]. For the priming phase, chromatin is open while the gene remains silenced. During the "coupled-on" or "coupled-off" phase, chromatin accessibility and gene expression increase or decrease together, respectively. Finally, the decoupled phase refers to chromatin accessibility decreasing while gene expression increases (or inverse, examples in Supplementary Figs. 8C and 8D). For example, priming and coupled-on phases have been observed for *CLU* in PRPCs (Fig. 5G). For PRPCs, primed, coupled-on, and coupled-off are more common than decoupled (Fig. 5H). Next, the percentage of priming time among genes identified in the PRPC module analysis was compared (modules from Fig. 2E). Module 1 comprises genes expressed in early PRPCs, while Module 3 contains genes expressed in late PRPCs; A significantly larger proportion of coupled-on time was observed among genes in Module 3 compared to genes from Module 1, indicating that at later stages, gene and chromatin accessibility undergo more concordant changes. (Fig. 5I, Supplementary Fig. 8E and 8G). The distribution of different phases was calculated for variable genes across different major classes (Fig. 5J). Interestingly, on average, major classes with less number of subclasses have shorter coupled-off time, such as rods (20.98%), and cones (29.77%), while major classes with more cell types have longer coupled-off time, such as ACs (49.07%), and BCs (35.15%).

## Comparison of Macular and Peripheral Developing Retina

Among mammals, one of the unique features of the simian primate retina is the development of the cone-rich fovea and macula, which differs significantly from the peripheral retina in cell composition. Previous studies in model organisms suggest RA is involved in foveal formation[47]. To examine if RA inhibition in macula is conserved in human foveal formation, the expression patterns of RA pathway related genes, including retinaldehyde dehydrogenases enzyme coding genes (*ALDH1A1*, *ALDH1A2*, and *ALDH1A3*), and RA-catabolizing enzyme coding genes (*CYP26A1*, *CYP26B1*, and *CYP26C1*) were examined (Fig. 6A and Supplementary Fig. 9). In addition, T-Box transcription factor 5 (*TBX5*) and ventral anterior homeobox 2 (*VAX2*), which are dorsoventral patterning TFs that regulate RA related enzymes[48], and two fibroblast growth factors (*FGF8*, and *FGF9*) were examined (Supplementary Fig. 9). Consistently, *ALDH1A1* and *ALDH1A3*, which are involved in RA synthesis, are enriched in peripheral PRPCs whereas *CYP26A1*, which functions in RA degradation, is highly expressed in macular PRPCs. Differently from *ALDH1A1* and *ALDH1A3*, *ALDH1A2* is enriched in the macula. Moreover, *CYP26B1*, and *CYP26C1* have little expression in either location. Moreover, in PRPCs, although *VAX2* expression is similar in the peripheral and macular PRPCs (Supplementary Fig. 9), the periphery had a stronger binding signal for the

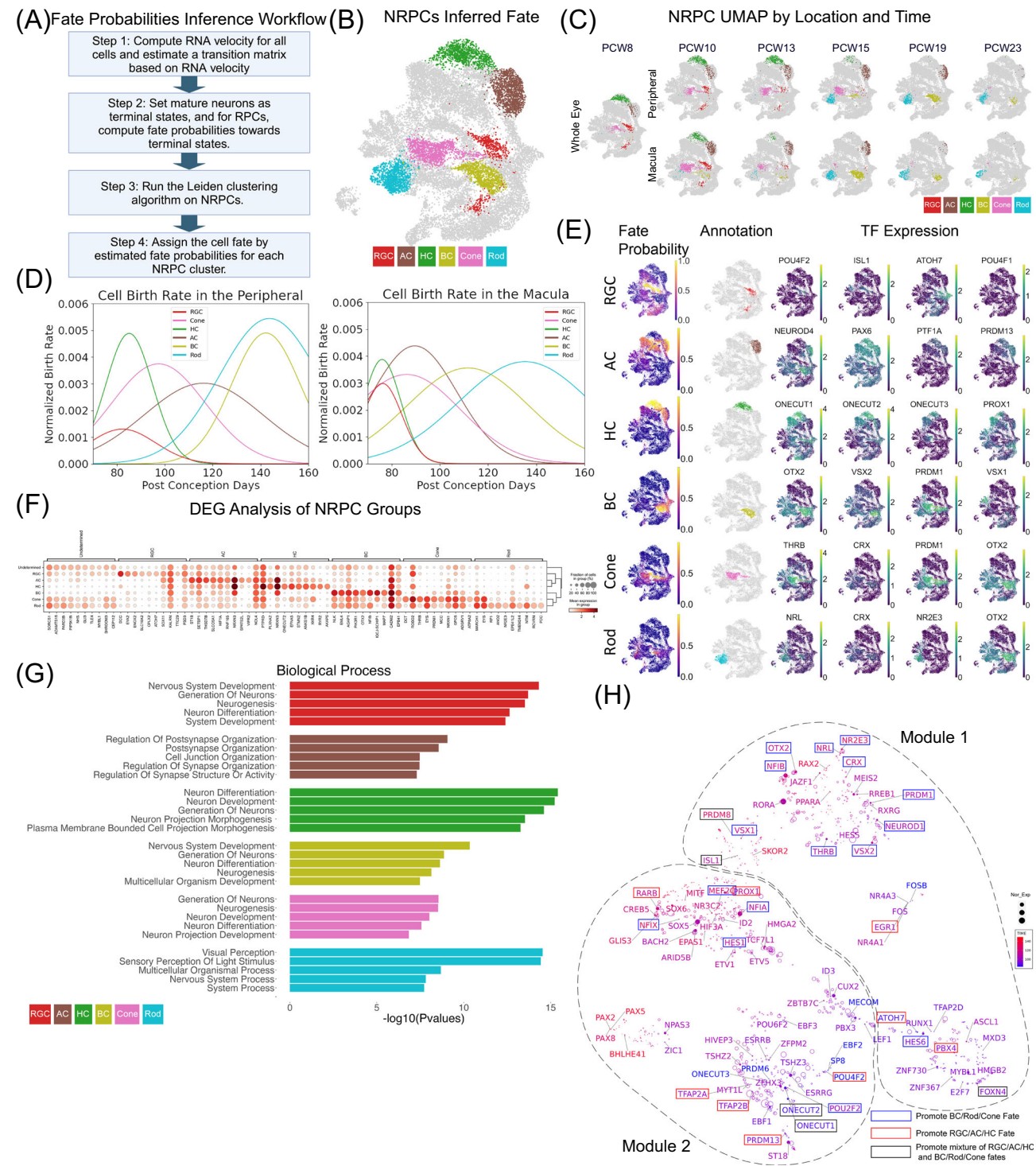

VAX2 motif (Supplementary Fig. 10B). *TBX5* has higher expression in peripheral PRPCs, but *TBX5* motif binding shows no differences between macula and periphery (Supplementary Fig. 10A), which suggests a different regulation of these two genes. Inconsistently, *FGF8* has been shown to be highly expressed in the high-acuity area in chick but is enriched in peripheral PRPCs in our data.

Subsequently, the temporal expression patterns of genes associated with RA were compared with previous publications. In both the macula and periphery, the expression of *ALDH1A1* and *ALDH1A3* demonstrates a decreasing trend over time[49] (Fig. 6B). Conversely, the expression of *ALDH1A2* exhibits an upward trend over time, consistent with prior findings. Notably, the expression levels of *ALDH1A2* are

consistently higher than those of *ALDH1A1* and *ALDH1A3* for most observed time points, deviating from previous observations. In particular, *TXB5* and *ALDH1A1* show similar gene expression patterns over time.

Next, the expressions of previously reported candidate genes associated with macular development[50] are examined. As expected, these genes have consistently exhibited higher expression levels in the macula throughout various developmental stages compared with periphery. (Fig. 6C).

To further investigate the difference of macular and peripheral retinal development, DEG analysis was performed after correcting for development timing variables (Supplementary Data 11). As a result, 47

**Fig. 3 | NRPCs with different fates show distinct gene expression patterns.**
**A** Diagram of the NRPCs' fate probability inference workflow. Cell fate probabilities for each type of neuron were estimated based on velocity, and cell fate was assigned after performing cell clustering. **B** UMAP of NRPCs colored by inferred fate. Cells for which cell fate cannot be determined are labeled in gray. The UMAP is specifically for NRPCs, distinguishing it from the global UMAP designed for all cells. **C** UMAP of NRPCs colored by inferred fate separated by region and time. **D** The cell birth rate was estimated for each NRPC group (grouped by inferred fate), within both the peripheral and macular regions. **E** UMAP of NRPCs colored by fate probability for each major class (first column), with annotated NRPCs for each fate (second column), and corresponding established TFs driving that fate (third to sixth columns). **F** Dot plot of differentially expressed genes for each NRPC group, with a different fate inferred. To identify differentially expressed genes, the two-side overestimates variance t-test[59] was applied to different fate groups among 21,087 cells. The Benjamini-Hochberg procedure was applied. **G** Gene ontology

analysis of differentially expressed genes for each NRPC group. To measure the significance of a functional term, the one-side hypergeometric test was used with the input gene list (100 genes in each module). Top 10 significant biological processes were plotted with adjusted p-values from Benjamini-Hochberg procedure. **H** GRN visualization of NRPC regulators. TFs were colored based on gene expression weighted time. The size of each dot represents the average normalized gene expression. UMAP was constructed based on the similarities of regulation effects on target genes. K-means clustering identified three clusters of TFs, and clustering information is depicted as a dashed line. The rectangle surrounding each TF indicates its confirmed impact on cell fate decisions, validated through loss-of-function animal experiments (Supplementary Data 8). A blue rectangle signifies that the TF promotes the BC/Rod/Cone fate, and a red rectangle indicates the promotion of the RGC/AC/HC fate. A black rectangle means the TF supports a mixture of two. **A** was created with BioRender.com.

**Table 1 | Summary of TFs in the NRPC GRN**

| Major Class | Matched | Unmatched | Unknown |
|---|---|---|---|
| AC | PRDM13 | HES6 (Rod), POU2F2 (Cone) | ID2, ST18 |
| BC | NFIB, OTX2, PRDM8, VSX1, VSX2 | NEUROD1 (Cone, Rod) | RORA, SKOR2 |
| Cone | FOXN4, PRDM1, THRB | | ID3, CUX2, FOS, FOSB, MEIS2, NPAS3, NR4A1, NR4A3, RUNX1, RXRG |
| HC | ONECUT1, ONECUT2, PROX1, TFAP2A, TFAP2B | | EBF1, ESRRB, ESRRG, HIVEP3, MYT1L, ONECUT3, PRDM6, TSHZ2, TSHZ3, ZFHX3, ZFPM2 |
| RGC | ATOH7, ISL1, PBX4, POU4F2 | EGR1 (AC, HC), ZIC1 (Rod) | E2F7, EBF2, EBF3, FOXM1, HES5, LEF1, MECOM, MXD3, MYBL1, PAX5, PBX3, POU6F2, SOX5, SP8, TFAP2D, ZNF367, ZNF730 |
| Rod | CRX, HMGA2, NR2E3, NRL | HES1 (Cone), MEF2C (Cone), NFIA (BC), NFIX (BC), RARB (RGC) | ARID5B, ASCL1, BACH2, BHLHE41, CREB5, EPAS1, ETV1, ETV5, GLIS3, HIF3A, JAZF1, MITF, NR3C2, PAX2, PAX8, PPARA, RAX2, RREB1, SOX6, TCF7L1, ZBTB7C |

First, TFs that promote MG differentiation were removed. The predicted functions of TFs are based on their normalized gene expression values in annotated NRPCs. TFs in the "Matched" column represent those with correctly predicted functions, validated through loss of function experiments using animal models. TFs in the "Unmatched" column indicate functions are mis-assigned. TFs in the "Unknown" column have unclear functions in the context of retina development.

genes are enriched in the macula and 177 genes are enriched in the periphery in PRPCs and distinct expression patterns for the top DEGs across the two locations were observed (Fig. 6D). With GO analysis, the term "cell adhesion" was identified within both macula and periphery enriched DEGs, but the two groups of "cell adhesion" related genes are totally different (Table 3 and Supplementary Data 12). For example, *CCN1*, *S100A10*, *MPZ* are enriched in the macula while *LRRN2*, *CYP1B1*, *DAPL1* are enriched in the periphery.

DEGs are also observed in late major classes between the macular and peripheral retina, with a total of 1163 genes enriched in the macula and 988 genes enriched in the periphery (Supplementary Data 11). Rods share most of the macular enriched DEGs and RGCs shared most of the peripheral enriched DEGs (Fig. 6E).

### Macular hypoplasia related gene expression
Foveal hypoplasia is a condition characterized by an incomplete development or complete absence of the fovea centralis with significant clinical and genetic heterogeneity. The underlying molecular pathways range from defects in pigmentation, achromatopsia, to early development[51].

We examined gene expression associated with typical and atypical foveal hypoplasia related conditions (Fig. 7A). Genes associated with ocular albinism (OA) and oculocutaneous albinism (OCA) show little expression in the retina, which is expected since those genes are characterized by pigmentation defects.

Furthermore, Hermansky–Pudlak syndrome (HPS) associated gene *AP3B1* is expressed in RPCs, and *LYST*, which is associated with CHS (Chediak–Higashsyndrome), shows high expression in developing BCs and photoreceptors only. Finally, most of the achromatopsia related genes showed high expression in cones, consistent with its

known characteristic of non-functioning cones. Hypoplasia related DEGs between macular and peripheral analysis reveals that only *CNGB3* and *PDE6H* exhibit significantly higher expression levels in macular cones compared to peripheral cones (Fig. 7B). Both *CNGB3* and *PDE6H* show a significant increase in gene expression levels after 90 days post conception in the macula and 115 days in the periphery (Fig. 7C). Furthermore, there is a consistent pattern of increased chromatin accessibility for both *CNGB3* and *PDE6H* in macular cones compared to their peripheral counterparts (Supplementary Fig. 11C). Peaks around *PDE6H* showed greater accessibility in the macula than periphery (Fig. 7E).

To test if complex eye diseases are related to specific major classes during development, GWAS related loci were examined (Fig. 7F and Supplementary Fig. 11E). Overall, 12 GWAS traits showed significant enrichment in 23 subclasses. As expected, photoreceptor cell measurement− including outer nuclear layer (ONL) thickness, inner segment (IS) thickness and outer segment (OS) thickness−associated loci were significantly enriched in cones, rods and their precursors. Loci associated with primary open-angle glaucoma (POAG) showed significant enrichment in NRPCs and MGs.

## Discussion
In this study, we present a comprehensive, high-resolution, dual-omic atlas of the developing human retina. We profiled over 220 K nuclei from 14 human embryos and fetuses, ranging from PCW 8 to 23, enabling the identification of more than 60 distinct cell classes and states throughout development. This atlas is comprehensive in three significant aspects. First, by employing dual-omic technology, this atlas concurrently captures gene expression and chromatin structure for each cell, avoiding potential errors arising from the computational

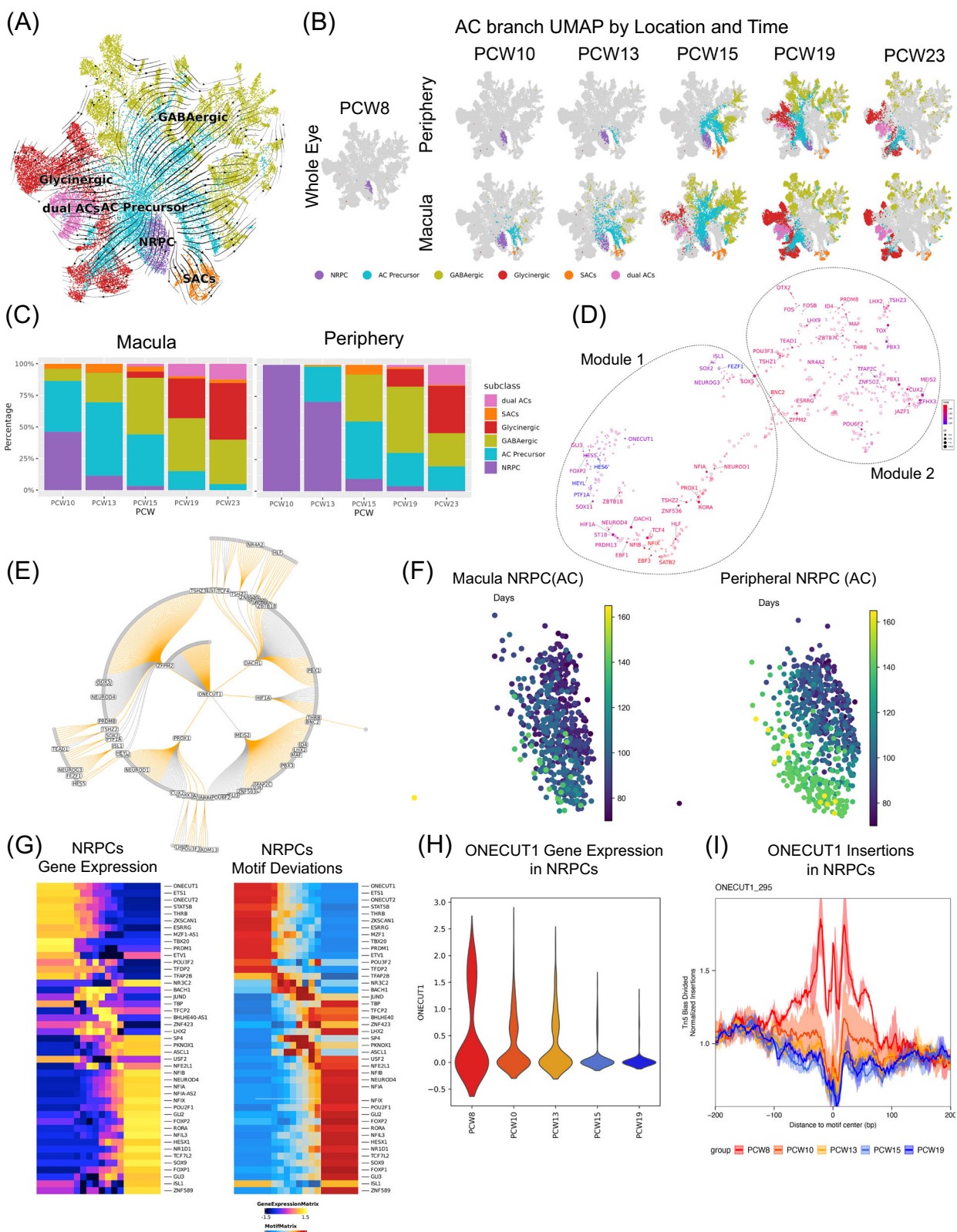

integration of these two modalities. Second, the human embryos and fetuses evenly span PCW 8–23, encompassing most of the pivotal developmental stages. Notably, a large portion of our data consists of progenitors and precursors, facilitating the identification of genes and pathways associated with cell fate determination. Moreover, given the extensive number of nuclei profiled, this atlas captures all major cell classes and the majority of subclasses and cell types in the retina.

Third, the dataset includes both the macula and periphery, enabling the investigation of spatial differences.

The inclusion of both macular and peripheral locations from each human embryos and fetuses eliminates any potential bias arising from individual variation. This unbiased comparison reveals several interesting observations about cell birth order and timing between the two regions.

**Fig. 4 | AC Subclass formed from specific groups of NRPCs at different times. A** Stream plot of velocities on the AC branch UMAP. UMAP was calculated on AC branch cells only, which is different from global UMAP. Cells were colored by subclass representing AC progenitors, AC precursors, SACs, GABAergic ACs, Glycinergic ACs, and dual ACs. **B** AC branch UMAP colored with subclasses separated by spatial location and time. **C** Proportions of AC subclasses in the macula and periphery at different time points. **D** GRN visualization of AC regulators. TFs were colored by gene expression weighted clock time. The size of each dot is the average gene expression value in log space. All TFs are labeled with text, while the target genes that are not TFs are left unlabeled. The UMAP was built on similarities of regulatory effects on target genes. Two modules were identified with K-Nearest Neighbors (KNN) algorithm. **E** GRN subgraph for *ONECUT1* in the AC branch, displaying first- and second-order ONECUT1 target genes. All TFs are labeled with text, while the target genes that are not TFs are left unlabeled. The edges are colored based on the TF regulatory interaction, with orange indicating positive regulation and gray indicating negative regulation. **F** UMAPs of AC progenitors separated by spatial locations and colored by days post-conception. **G** Heatmap for gene expression and motif deviations of AC progenitors. Cells were arranged according to the pseudotime inferred from the ATAC-seq trajectory. 45 features were selected based on the Pearson correlation between gene expression and motif deviation, with a threshold set greater than 0.3. **H** ONECUT1 gene expression in AC progenitors over time. **I** Tn5 bias-adjusted TF footprints for ONECUT1 motifs. Lines are colored by sample PCW groups.

Due to the central-to-periphery development of the retina, macular samples consistently exhibit a higher percentage of differentiated cells. Consistent with previous immunolabeling results from[17], MGs and a few BCs can be identified as early as 59 days in whole eye retina, as shown in Fig. 1G (Supplementary Fig. 1I). Although these cells can be detected early, it takes much longer for them to reach their maximum proportions. Based on abundance, the calculated cell birth order in the macula closely corresponds to that in the periphery and agrees with prior publications[51]. One exception, shown in Fig. 1G, is that a significant number of rods appear earlier than an equal amount of BCs in the periphery, while the inverse is observed in the macula. Further experiments are necessary to confirm this observation. Although all major classes appear in the macula before the periphery, the time delays vary depending on the major class. Early-born major classes have similar timings when comparing the macula and periphery. Conversely, late-born major classes like BC and MG in peripheral areas exhibit older ages compared to those in the macula. When comparing the cell percentage between the macula and periphery, the percentage of rods is greater in the periphery compared to the macula from PCW 15 (Supplementary Fig. 4A).

It has been demonstrated that RPCs undergo dynamic transitions during development, altering their potential to differentiate into various major classes. However, it remains unclear whether cell state transitions occur continuously or discretely. Given the extensive number of RPCs profiled in the current dataset, we have the capability to produce a high-resolution transcriptomic profile detailing changes throughout development. Overall, the PRPCs formed a continuous cluster without clear discrete transitions. After removing cell cycle related genes, both macular and peripheral PRPCs gradually shift in the same direction on the UMAP and this shift is mainly driven by developmental time (Fig. 2A). This suggests that, although early PRPCs and late PRPCs have distinct gene expression profiles, they form in a gradual and continuous manner.

The potency of PRPCs influences the birth order and proportion of various retinal neuron types. This is likely driven by gradual changes in gene expression and epigenetics. However, the precise molecular pathways driving the transition remain unclear. Analysis of the 55,000 PRPCs profiled reveals multiple gene clusters with changing expression patterns during development. Interestingly, while PRPCs seem to exhibit a smooth transition, the expression of certain gene modules demonstrates more pronounced differences between early and late PRPC stages (Fig. 2E). based on gene expression, we observed that the transition occurs for some progenitors in the macula as early as PCW 8, and continues to PCW 15, and in the periphery around PCW 19 (Fig. 2A). Coinciding with the transition time window, significant increases in the expression of the ribosomal protein gene pathway were observed, including 46 ribosomal protein L genes and 31 ribosomal protein S genes (in PRPC Gene Module 2), which are associated with protein biogenesis. Consistent with these findings, previous publications have reported *Rps7* disruption results in eye malfunction in mouse[52], and *Rpl24* can alter cell cycle and cell fate determination[53].

We further looked into the chromatin regulation during the transition. First, early expressed genes have a shorter coupled-on time compared to late ones, indicating that after the transition, gene expression and chromatin changes are more concordant. Second, we found that ribosomal protein genes show delayed chromatin changes compared to gene expression. For example, before transition, *RPS14* expression and chromatin accessibility increased concordantly (Supplementary Fig. 8F). However, after the transition, unspliced and spliced *RPS14* levels decrease, but the chromatin accessibility level still increases for a period of time.

Unlike the gradual transition observed in the PRPC state, the differentiation of individual major classes appears to follow a more distinct pattern. Through trajectory analysis, we can readily identify the progenitor cells for each major class and subclass throughout development. Intriguingly, even for subclasses within the same major class, we identified distinct sets of progenitors. For instance, ACs can be divided into three subclasses, with SACs differentiating first, followed by GABAergic ACs, and lastly, Glycinergic ACs. Through trajectory and velocity analysis, we observed that the progenitors of SACs, GABAergic, and Glycinergic ACs appear in the corresponding order and form distinct subclusters (Fig. 4B), indicating a hierarchical cell development model (Supplementary Fig. 7K). Using *ONECUT1* as an example, we showed that differences in gene expression and chromatin accessibility are already present at the progenitor stage, establishing distinct clusters that predispose cells to specific subsequent fates.

By integrating transcriptomic and chromatin profiling data, we systematically identified GRNs linked to specific cell states and their transitions. Notably, our identified GRNs encompass numerous known retinal development-related genes and gene modules. For example, among TFs, 61 for ACs, 104 for BCs, 116 for Rods, 114 for cones, 174 for RGCs, and 103 for HCs (Supplementary Data 6). Future studies that incorporate complementary epigenetic signatures and Hi-C data can provide more detailed insights into the GRNs.

In addition to the recognized genes and gene modules, our analysis has unveiled a plethora of candidate genes that may play pivotal roles in retinal development. Notably, a substantial group of zinc finger protein-related TFs has been identified, yet their specific functions in retinal development remain largely unexplored. Intriguingly, our findings reveal that, even within the same lineage during development, significantly different regulatory mechanisms may exist for each lineage. For instance, Supplementary Fig. 7A illustrates distinct TFs and target genes identified for AC progenitors, AC precursors, and mature ACs. This suggests that cell fate specification and differentiation are achieved through diverse regulation involving different key players. A similar pattern is observed for HCs and cones, as demonstrated in Supplementary Fig. 6.

These discoveries emphasize the complexity of retinal development, showing the different regulatory networks governing the differentiation and specification of the same cell lineage. The identified candidate genes, especially the zinc finger protein-related TFs, present promising avenues for further research to unravel their specific roles in the intricate landscape of retinal development.

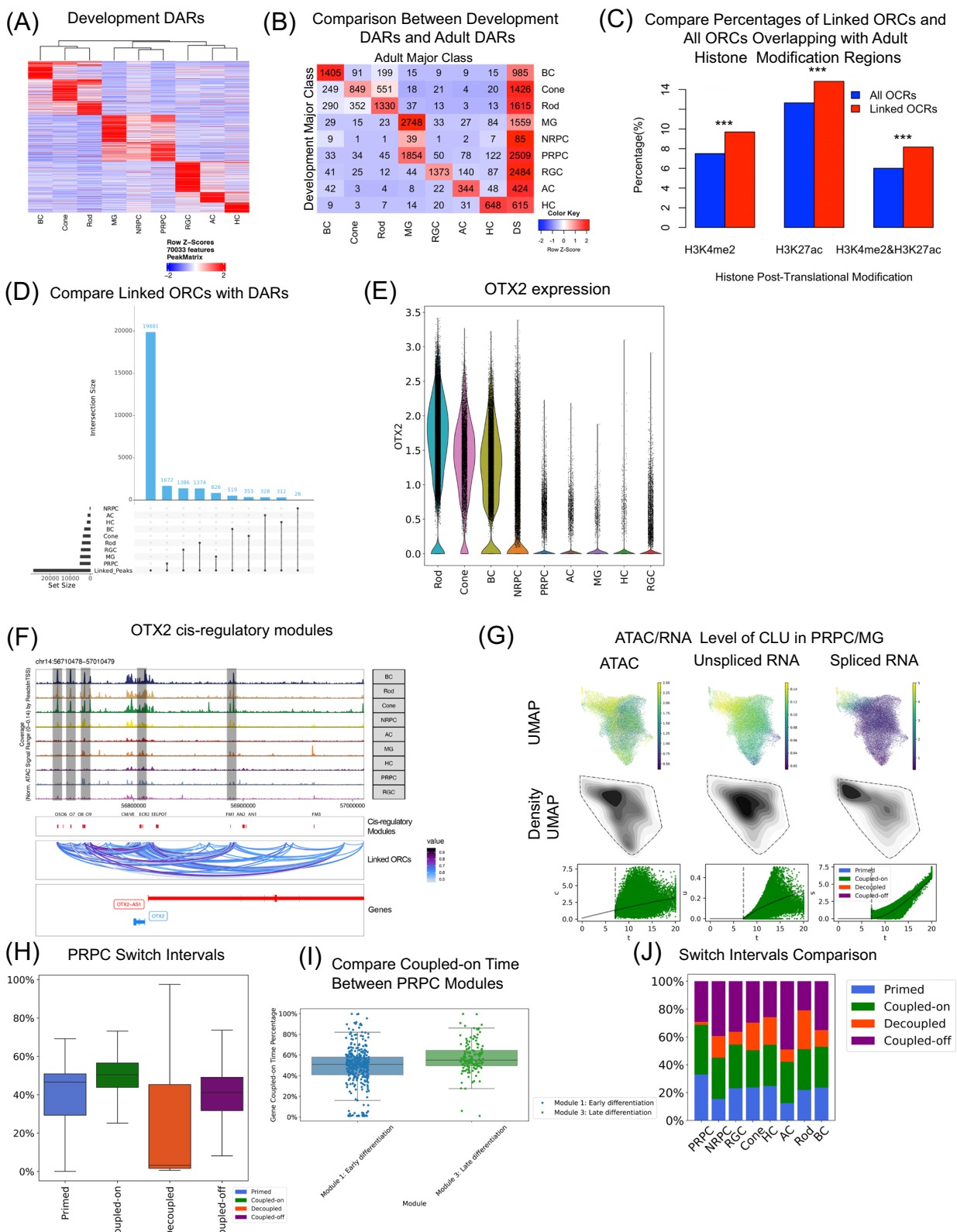

Our dataset reveals an elevated activation of the RA signaling pathway in the peripheral retina compared to the macula. We identified several RA pathway genes that exhibit differences between humans and chickens. For example, *CYP26A1* nor *CYP26C1* was expressed in the human retina up to PCW 9. Our data shows high expression of *CYP26A1* in late macular PRPCs at around PCW 20, but no expression of *CYP26C1* at any time. In addition, the expression pattern of *FGF8* in PRPC diverges. Rather than being predominantly expressed in the macula, *FGF8* and *FGF9* exhibit higher expression in peripheral PRPCs. This suggests a potential variation in the function of the *FGF* pathway between humans and chickens in retinal patterning.

**Fig. 5 | Different epigenome-transcriptome interaction patterns regulate development. A** Heatmap showing DARs for each major class. Marker peaks were identified using the one-side Wilcoxon rank-sum test using the Benjamini-Hochberg correction, applying a threshold of false discovery rate <0.01 and a Log2 Fold Change >1. **B** Heatmap illustrating the count of identified developmental retinal DARs overlapping with adult retinal DARs. DS, representing "developmental-specific" DARs that do not match with any adult DARs. **C** The bar plot represents a comparative analysis of two distinct categories of adult histone modification regions. Two-sided two-sample tests for equality of proportions with continuity correction were performed to compare each pair of proportions. The Benjamini-Hochberg procedure was applied. Adjusted $p$-values are smaller than 2.2e-16. **D** Upset plot comparing major-class marker peaks and linked peaks. The set size represents the number of peaks within each category. The intersection size represents the number of peaks for the corresponding combination. **E** *OTX2* normalized gene expression violin plot by major class. **F** Genome browser displaying chromatin accessibility around cis-regulatory modules (identified in the mouse by

Chan C. S et al.[45]) located near *OTX2*. The cis-regulatory modules are denoted by red vertical bars in the "cis-regulatory modules" panel. The "Linked ORCs" section illustrates open chromatin regions linked to genes. The transparent black-box-labeled peak indicates the overlap between identified linked ORCs and reported cis-regulatory modules. **G** UMAPs colored by levels of chromatin openness, unspliced mRNA and spliced mRNA for *CLU* in PRPCs. Those levels were plotted in density plot and line chart against gene time. **H** Box plots summarizing the lengths of each of the four cell states across all fitted genes for PRPCs. In total, 1790 genes were fitted among 52,479 cells. The bars represent the minimum, median, and maximum and the box spans the 25th to 75th percentile. (same for (**I**)). **I** Box plot comparing the percentage of coupled-on time within gene time between PRPC modules (Fig. 2E). A two-side Welch two-sample t-test was applied to compare coupled-on time percentages between 378 module 1 genes and 140 module 3 genes. The adjusted $p$-value is 1.091e-07 with Benjamini-Hochberg applied. **J** Bar plot summarizing the percentage of the four cell states of all major classes.

**Table 2 | Functions of cis-regulatory regions of RGC differentially accessed regions (DARs)**

| RGC DAR Type | Terms | Binom FDR Q-Val |
|---|---|---|
| Development Specific RGC DARs | Retinal ganglion cell axon guidance | 1.46E-06 |
| Development Specific RGC DARs | Cell migration in hindbrain | 6.23E-06 |
| Development Specific RGC DARs | Radial glia guided migration of Purkinje cell | 5.83E-05 |
| Development Specific RGC DARs | Transmission of nerve impulse | 1.10E-04 |
| Development Specific RGC DARs | Nerve development | 1.16E-04 |
| Shared RGC DARs with Adult | Nerve development | 1.20E-07 |
| Shared RGC DARs with Adult | Innervation | 6.34E-07 |
| Shared RGC DARs with Adult | Negative regulation of neuron death | 9.98E-07 |
| Shared RGC DARs with Adult | Trigeminal nerve development | 1.37E-06 |
| Shared RGC DARs with Adult | Positive regulation of cyclase activity | 3.41E-06 |

"Development Specific RGC DARs" are the DARs found in developmental data but not overlap with Adult DARs; "Shared RGC DARs with Adult" are DARs that can be found in developmental data that can also be found in adult data.

## Methods

### Ethics and tissue acquisition

This research complies with tenets of the declaration of Helsinki. The study was reviewed and approved by the UC Davis Institutional Review Board (IRB) (IRB ID: 903054-1). The use of discarded de-identified human fetal retinal tissue was approved by the UC Davis Stem Cell Research Oversight Committee (SCRO protocol#1171, initial approval 12/16/2019). Human ocular tissues were obtained from discarded de-identified fetal waste from elective abortions. The patients from whom the tissues were derived were informed and freely agreed for them to be used for research purposes. Only tissues from women agreeing to donate it for scientific research were used in this study. Their agreement for research was documented in the medical chart. Patients were not compensated.

### Human developmental sample collection

The human samples from PCW 10 to PCW 23 were collected from 12 individuals (with IDs 2 and IDs from 4 to 14 in Supplementary Data 1) in this study from the UC Davis Eye Center. All the samples were collected within 6 h post-mortem, according to the protocol[54]. The macula and peripheral retina samples were collected with a 2 mm disposable biopsy punch and flash-frozen by liquid nitrogen. Samples were then stored at −80 °C. All tissues were de-identified under HIPAA Privacy Rules. Gender was determined by the relative gene expressions of *XIST* and *DDX3Y*. Only tissues from women agreeing to donate it for scientific research were used in this study. Their agreement for research was documented in the medical chart.

### Nuclei isolation and sorting

Nuclei were isolated by WheatonTM Dounce Tissue Grinder in prechilled freshmade RNase-free lysis buffer (made with 10 mM Tris-HCl, 10 mM NaCl, 3 mM MgCl$_2$, 0.02% NP40, 1% BSA, 1 mM DTT, and 1U/ul RNAse inhibitor). Being triturated by the loose and tight dounces, tissue structure was broken and homogenized. Isolated nuclei were washed twice by fresh-made wash buffer (10 mM Tris-HCl, 10 mM NaCl, 3 mM MgCl$_2$, 1% BSA, 1 mM DTT, 1U/ul RNAse inhibitor) for 5 minutes at 4 °C, 500 g. The pellet was then re-suspended in a diluted 1X Nuclei Buffer (10X Genomics, 2000153/200).

### Single-nuclei dual-omic sequencing

All single-nuclei dual-omics-sequencing in this study was performed at the Single Cell Genomics Core at Baylor College of Medicine following the library construction protocol CG000338 from 10x Genomics. Single-nuclei cDNA library preparation and sequencing were performed following the manufacturer's protocols (https://www.10xgenomics.com). The single-nuclei suspension was introduced into a Chromium controller in order to generate single-cell GEMS (Gel Beads-In-Emulsions) for the subsequent reaction. The snRNA-seq and snATAC-seq library were prepared with Chromium Next GEM Single Cell Dualomic GEM kit (10x Genomics). The library was then sequenced on Illumina Novaseq 6000 (https://www.illumina.com).

### Obtain additional public datasets

Dual-omic sequencing data (in bam format) of three additional developmental retina samples (with IDs 1 and 3) were retrieved from the National Center for Biotechnology Information, specifically from project GSE246169 [http://www.ncbi.nlm.nih.gov/geo/query/acc.cgi?acc=GSE246169]. The samples were processed following the experimental procedures mentioned in ref. 17. In GSE246169, sample 1 corresponds to Day 59 Human Fetal Retina, while sample 2 represents Day 76 Human Fetal Retina Center, and sample 3 corresponds to Day 76 Human Fetal Retina Peripheral in Supplementary Data 1.

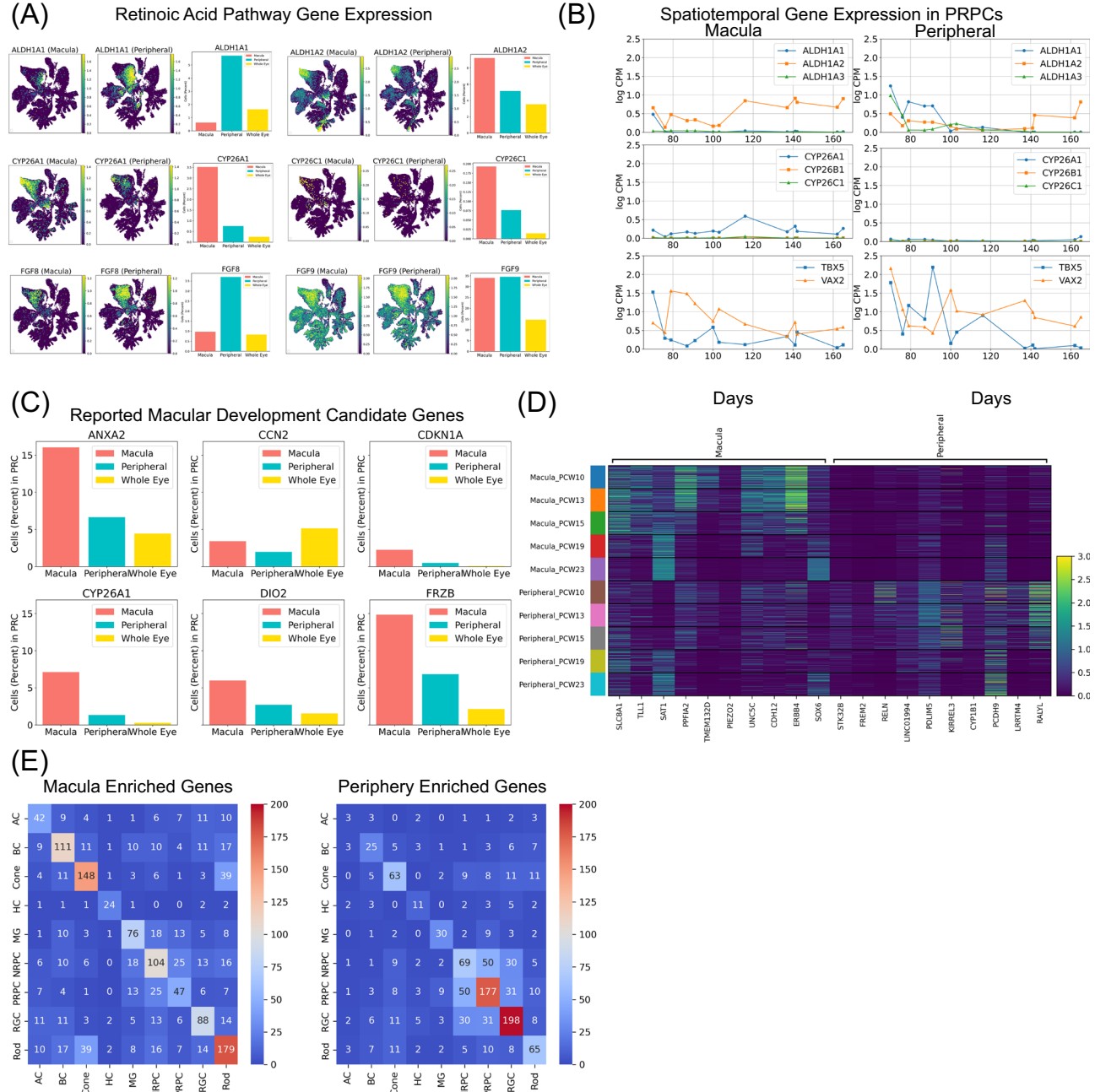

**Fig. 6 | Gene expression comparison of macular and peripheral retina. A** Marker gene expression of the RA pathway, including *ALDH1A1*, *ALDH1A2*, *CYP26A1*, *CYP26C1*, *FGF8*, and *FGF9*. For each gene, both UMAPs (Macula and Periphery information labeled in the caption) and bar charts displaying the proportion of cells with non-zero gene expressions were plotted. **B** Line chart showing the natural log-transformed counts per million of *TBX5*, *VAX2*, *ALDH1A1*, *ALDH1A2*, *ALDH1A3*, *CYP26A1*, *CYP26B1*, and *CYP26C1* in macula and periphery over time. **C** Bar charts displaying the proportion of cells with non-zero gene expressions for previously reported macula development candidate genes[50]. **D** Heatmap of top 10 PRPC DEGs identified in the macula and periphery. The first half of genes are enriched in the macula and the second half of genes are enriched in the periphery. **E** Heatmap showing the number of DEGs identified among all major classes. The ones on the diagonal are genes identified as DEGs exclusively in one major class, while the rest are overlapped DEGs identified in more than one major class.

## Quality Control Overview

In total, Quality Control comprises 4 filtering steps. For each step, only cells that have passed all previous steps will be used. Here is an overview of the quality controls: Filter 1: Filter cells by the number of features per cell and the percentage of mitochondrial counts. Filter 2: Doublet identification based on RNA-seq. Filter 3: Doublet identification based on ATAC-seq. Filter 4: Filter by major class annotation. Each filter is explained in detail in the following sections.

## RNA-seq Data Quality Control

Raw sequencing data in FASTQ format were processed using Cell Ranger ARC software (either cellranger-arc-1.0.0 or cellranger-arc-2.0.0). This software facilitated the identification of cells, alignment to the GRCh38 (GENCODE v32/Ensembl 98) human reference genome, and the generation of gene-barcode matrices (see Supplementary Data 1). Seurat (v4.2.0)[55] was employed for initial filtering. In each sample, features expressed in ten or fewer cells were excluded from downstream analysis. Subsequently, quality control procedures

**Table 3 | PRPC Enriched Biological Processes**

| PRPC Enriched GO | Region | Adjusted P-Value | Intersections |
|---|---|---|---|
| Postsynaptic Intermediate Filament Cytoskeleton | Macula | $5.494 \times 10-3$ | NEFM, NEFL |
| Cell Adhesion | Macula | $1.180 \times 10-2$ | ADAMTSL1, CDH12, UNC5C, PCDH8 |
| Collagen Fibril Organization | Macula | $1.767 \times 10-2$ | LUM, COL1A1, ANXA2, TLL1 |
| Collagen-containing Extracellular Matrix | Macula | $4.428 \times 10-2$ | LUM, COL1A1, CCN1, ANXA2, S100A10 |
| Cell Morphogenesis | Periphery | $5.085 \times 10-13$ | LRATD1, ANO1, ROBO2, PDLIM5, CDH6 |
| Cell Adhesion | Periphery | $5.075 \times 10-6$ | DAPL1, ROBO2, PDLIM5, CDH6, DCC |
| Cell Motility | Periphery | $2.501 \times 10-3$ | LRATD1, DCC, PRAG1, KIRREL3, TBX5 |
| Axon Guidance | Periphery | $6.991 \times 10-5$ | ROBO2, DCC, ANOS1, EFNA5, EPHA6, LHX9 |

Gene ontology analysis of DEGs enriched in PRPCs in the macula and periphery. The one-sided hypergeometric test was used to measure the significance of a functional term in the input gene list. The adjusted p-values were calculated from Benjamini-Hochberg procedure.

removed cells with fewer than 500 genes or more than 10,000 genes identified, as well as cells with greater than 10% mitochondrial content. The number of cells remaining after this filtering is provided in Supplementary Data 2, under the column "Pass.Filter.1".

Subsequently, doublets were inferred among all filtered cells, and DoubletFinder[56] was utilized to identify and remove these doublets. When employing DoubletFinder, the neighborhood size (Pk values) for each sample was determined based on the mean-variance normalized bimodality coefficient (BCmvn) score. Furthermore, during the parameter selection process, a maximum of 7.5% of doublets were allowed to prevent excessive data loss. The count of cells remaining after this filtering step is documented in Supplementary Data 2, under the column "Pass.Filter.2". Only cells passed processing were used for ATAC-seq analysis.

### ATAC-seq data quality control
ATAC-seq data were processed with Analysis of Regulatory Chromatin in R (ArchR)[57]. Cells with fewer than 1000 unique nuclear fragments or a signal to background ratio of less than 4 were removed.

Doublets were identified and removed based on doublet scores inferred from ArchR using the function "addDoubletScores()" with all default parameters. The count of cells remaining after this filtering step is documented in Supplementary Data 2, under the column "Pass.Filter.3".

Only cells passed both RNA-seq and ATAC-seq processing were used for downstream analysis.

### RNA-seq data integration and UMAP
For RNA-seq data integration, we first merged all filtered cells from all samples. Following this, UMAP and data integration for all cells were executed using scvi-tools (V1.0.3)[58].

In details, for feature selection, we selected the top 10,000 highly variable genes using Scanpy[59] "highly_variable_genes()" function with flavor set to "seurat_v3". Using scvi.model.SCVI(), we constructed the latent embedding, setting parameters to "n_layers=2", "n_latent=30", and "gene_likelihood = nb" (negative binomial). Afterward, we established a neighborhood graph and carried out Leiden clustering based on these latent embeddings.

### ATAC-seq UMAP
The UMAP for ATAC-seq data was calculated using the ArchR function "addUMAP()". Initially, iterative latent semantic indexing was computed for dimension reduction. Subsequently, to reduce sample-wise batch effect, "addHarmony()" was applied to eliminate any dimensions that had a Pearson correlation greater than 0.3 with the sequencing depth.

### Group samples based on post conception weeks (PCWs)
Samples were categorized into six PCW groups: PCW 8, PCW 10, PCW 13, PCW 15, PCW 19, and PCW 23. The allocation of samples to these groups was based on their sample post-conception weeks. Minor adjustments were additionally made to ensure that each PCW group contained approximately the same number of cells. The detailed results of this grouping can be viewed in Supplementary Data 1.

### Major class annotation
The overall major class annotation workflow diagram can be found at Supplementary Fig. 1A. In summary, major cell classes were annotated through reference mapping using scvi-tools (V1.0.3) and scArches[60] with default parameters. This process relied on an internally annotated adult dataset obtained from[2] (shared in https://zenodo.org/uploads/10866341). Specifically, the in-house human adult retina reference was trained to build an annotation model using a standard pipeline. Subsequently, the developmental data were updated to the model with the same adult latent space. UMAPs of developmental and adult data are provided in Supplementary Fig. 1B and 1C.

Cells annotated as retinal pigment epithelium (RPEs) or microglia were excluded from the analysis. The count of cells remaining after this filtering step is documented in Supplementary Data 2 under the column "Pass.Filter.4".

With the initial adult reference annotation set, both RPCs and MGs were identified as MGs (Supplementary Fig. 1D). To differentiate between RPCs and MGs, we relied on clustering and recognized marker genes for MG, namely *SLC1A3*, *RLBP1*, *SLN*, *SOX2*, *NFIA*, *CRYM*, *CLU*, and *LINC00461*, as outlined in a prior study[27]. UMAPs visualizing the expression of these genes are provided in Supplementary Fig. 1E. Within the RPC group, more detailed manual annotations were made between NRPCs and PRPCs, based on the expressions of established markers[27] in each identified cluster (see Supplementary Fig. 1E). The final annotation of RPCs can be found in Supplementary Fig. 1G. Annotation was validated by inferred cell cycle with tricycle[61] using all default parameters. For more validation, gene expression for each finalized major class is shown as a heatmap using established markers[27] in Supplementary Fig. 1H, with a corresponding gene score heatmap presented in the same panel.

### Subclass and cell type annotation
During major class annotation, the adult reference used only has major class information without subclass labels. So, within each major cell class, subclasses were identified using another adult reference data from[62]. The annotation was performed through a combination of clustering and manual annotation (Supplementary Fig. 2A). Adult data was obtained from the Human Cell Atlas Data Portal at HCA Data Portal with access ID 9c20a245-f2c0-43ae-82c9-2232ec6b594f. Next, both adult and developmental cells within each major cell class were integrated and clustered. Subsequently, the adult cell labels were manually transferred to developmental data based on the clustering results. Within each lineage, cells exhibiting distinct gene expression profiles compared to any adult subtypes were assigned corresponding precursor labels. Cell subclass annotations were then validated by

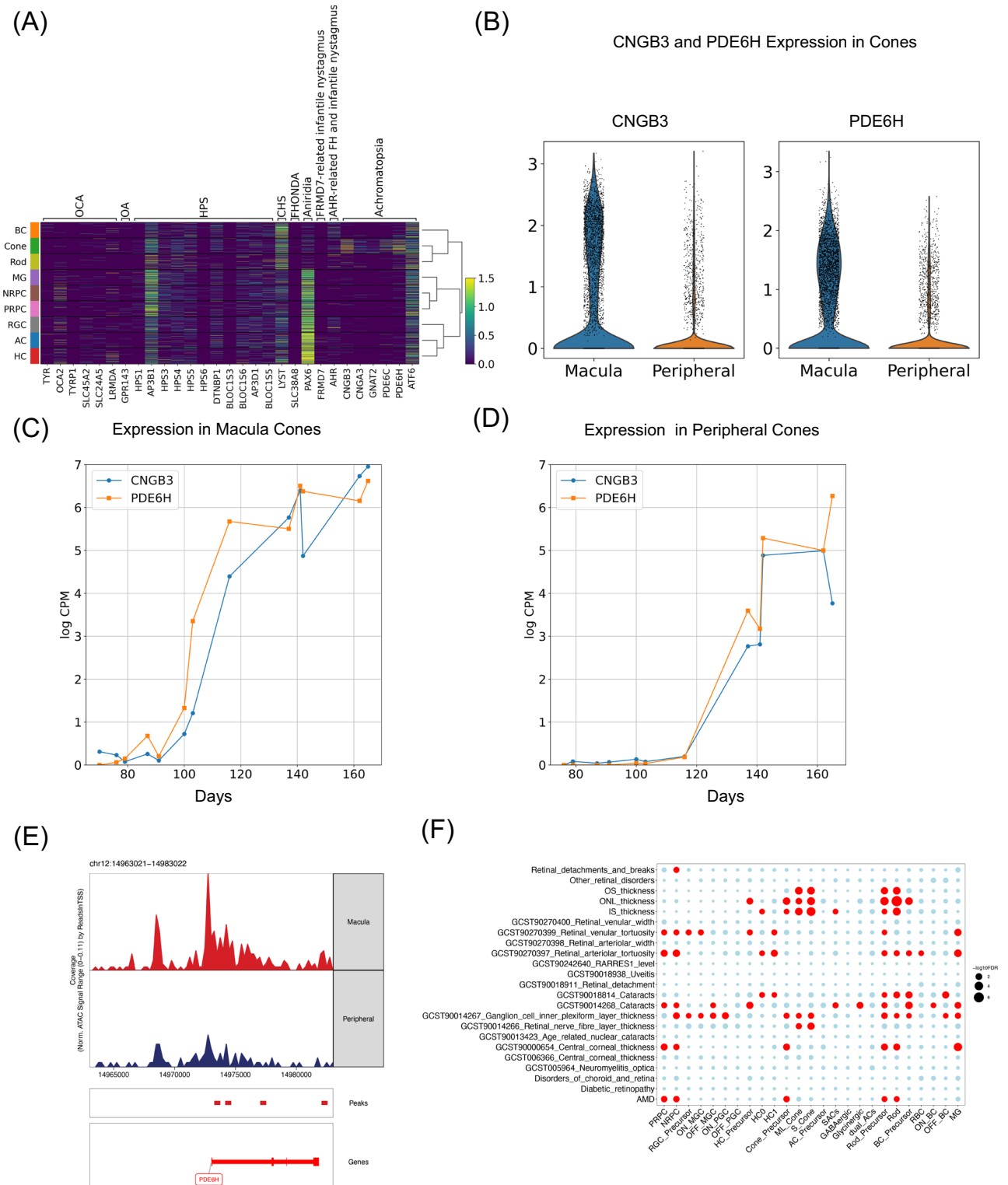

**Fig. 7 | Comparison of macular hypoplasia related gene expression in macula and periphery. A** Heatmap of genes associated with typical and atypical foveal hypoplasia. OCA (oculocutaneous albinism); OA (ocular albinism); HPS (Hermansky–Pudlak syndrome); CHS (Chediak–Higashi syndrome); FHONDA (foveal hypoplasia, optic nerve decussation defects and anterior segment dysgenesis). **B** Violin plot of *CNGB3* and *PDE6H* expression in cones, colored by macula and periphery. **C** Line chart of *CNGB3* and *PDE6H* log-transformed counts per million in the macular cones. **D** Line chart of CNGB3 and PDE6H log-transformed

counts per million in the peripheral cones. **E** Peak accessibility around *PDE6H* in the macular and peripheral cones. **F** The subclass enrichment of 23 eye-related GWAS traits based on gene expression from snRNA-seq data. AMD (age-related macular degeneration); IS (inner segment); ONL (outer nuclear layer); OS (outer segment). To test GWAS traits enrichment, a two-sided F-test was applied to compute the *p*-values. The Benjamini-Hochberg procedure was applied. Significant enrichment is highlighted in red (FDR < 0.05).

established marker gene expression. The details of the whole annotation process can be found at Supplementary Figs. 2A–G. The final subclass annotation UMAP can be found at Supplementary Fig. 2H. The annotation was further validated using marker gene expressions, as shown in Supplementary Fig. 3.

## Gene scores computation

A gene score was calculated for each cell at the gene level to measure the degree of chromatin accessibility. This was achieved by summarizing peaks near the gene's transcription start site, gene body, as well as at the promoter and distal regulatory elements, and applying appropriate weighting. In this study, we calculated gene scores to quantify the openness of each gene using ATAC-seq data with ArchR[57]. To estimate gene scores, we utilized ArchR with all default parameters, employing the "addGeneScoreMatrix()" function. During this process, gene scores were calculated through a weighted sum of accessibility within the gene body, as well as at the promoter and distal regulatory elements.

## Gene scores imputation

After gene score was computed, to enhance the visualization of gene scores, the "addImputeWeights()" function in ArchR[57] was employed to impute gene scores, utilizing default parameters.

## Differential gene expression analysis

Differential Gene Expression Analysis can be categorized into two main types. The first type involves comparing gene expression among major classes, subclasses, or distinct cell types. The second type focuses on examining gene expression differences between specific locations, such as the macula and the periphery. For the second type, we first accounted for and removed any confounding factors using regression analysis.

Differential gene expression analysis was performed among major classes, subclasses, or cell types using the Scanpy[59] built-in function "scanpy.tl.rank_genes_groups()". The gene expression count matrix was preprocessed with "scanpy.pp.normalize_per_cell()" and "scanpy.pp.log1p()" for normalization and natural log transformation, respectively. The analysis was then tested using the method = "t_test_overestim_var", which stands for overestimating the two-side variance in each group t-test. P-values were corrected using the Benjamini-Hochberg method, specifying corr_method = "benjaminihochberg".

Differential Gene Expression Analysis Between the Macula and the Peripheral Retina We applied the "Differential expression analysis" pipeline in Monocle3[63–67] for differential gene expression analysis between macula and periphery (Supplementary Fig. 9A). Monocle3 tests whether each coefficient differs significantly from zero under the two-side Wald test. Initially, cells were categorized based on their lineage into nine distinct groups: NRPCs, PRPCs, MGs, rods, cones, BCs, ACs, RGCs, and HCs. The sample size for each test is the number of cells in each major class. Subsequently, we began by fitting a regression model using the function "fit_models()", with "Location" (either "Macula" or "Periphery") as the sole explanatory variable. For determining differentially expressed genes (DEGs), we set our criteria to a "q_value of less than 0.01 and an absolute normalized effect greater than 1. Among these candidates, only genes present in over 2000 cells were considered. Next, we then employed a negative binomial model to account for gene expression variations. Our full model incorporated both "Location (categorical characters)" and "Age (Days, which are numeric integers)", while the reduced model was based solely on "Age". A likelihood ratio test was subsequently conducted to contrast the full and reduced models. Genes that met the criteria of "q_value < 0.01" were identified. Our final set of DEGs was discerned by finding the overlap between DEGs from the regression model and those from the likelihood ratio test.

## Maturation score calculation

For each major class, the maturation score was defined as its gene expression similarity to its corresponding type in adult cells. To quantify this similarity, the Pearson correlation was calculated. In detail, first, developmental cells and adult cells from the same major class were extracted to calculate the top 50 principal component vectors. Vectors were grouped by adults and different developmental stages and then averaged among all cells within each group. Subsequently, Pearson correlation was calculated between the averaged vector from adults and the averaged vector from different developmental stages to represent the maturation score.

## Cell birth rate estimation

To estimate the cell birth rate, which quantifies the percentage of births for each major class at specific time points, the following approach was applied. To counteract the effects of sample size variations, random sampling of 20,000 cells was performed for each sample. Following this, the mean and standard deviation of the sample age for cells were calculated. Using these values, a Gaussian kernel was fitted to create a distribution plot. Finally, to determine the overall proportion of each major class, their proportions were calculated. Subsequently, this proportion (a constant) was multiplied by the estimated Gaussian distribution. This ensured that the area under the curve of each Gaussian distribution accurately represented the proportion of the corresponding major class.

## Dual-omic dynamical modeling

Other than latent time inference, dual-omic dynamical modeling fitted by MultiVelo[46] can identify epigenome-transcriptome interactions. With MultiVelo, differential equations were used to model gene expression from RNA-seq and epigenetic data from ATAC-seq. The same model was trained and used before in "Latent Time Inference" in Methods. The top 2000 highly variable genes were used to each cell major class to train each model.

MultiVelo incorporates both RNA-seq and ATAC-seq data for velocity analysis and latent time inference. To get started, both RNA-seq and ATAC-seq data need to be prepared. For RNA-seq preparation: The counts of introns and exons were determined using veloctyo[68]. For ATAC-seq preparation: Cells with counts less than 2000 or greater than 50,000 were filtered out. MultiVelo then aggregated peaks, drawing from the annotated enhancers and promoters provided by the 10x CellRanger ARC output. These aggregated peaks underwent normalization and were smoothed by neighboring values to infer velocity. Subsequently, cell lineage was predicted focusing on velocities of the top 2000 highly variable genes. Then, for each branch, the dynamic model was estimated with "recover_dynamics_chrom()" function. To identify when the cell stage switch happened, scVelo[69] time inversion method was applied.

After fitting the dynamic model, MultiVelo can identify genes that are temporarily out of sync.

## Latent time inference

The latent time acts as a cell's intrinsic timer, estimating the cell's degree of differentiation during development or how much time has passed since differentiation began. It aggregates the time assignments per gene (gene time), calculated within scVelo's dynamical model[69], onto a global scale that accurately approximates the internal clock of individual cells. During development, within a lineage, we used latent time to sort cells from early to late. In this study, latent time was inferred with MultiVelo[46], a dual-omic single-cell velocity inference model. The velocity stream and inferred latent time were determined using MultiVelo's built-in functions, specifically "recover_dynamics_chrom()" and "latent_time()", with default parameters applied.

### PRPCs latent time comparison among PCWs and location

To verify the accuracy of the inferred latent time in representing the degree of development, we conducted a correlation analysis between PCWs (in numeric integer format) and latent time for all cells. Pearson's product-moment correlation coefficient was employed, with the null hypothesis positing that the correlation between PCWs and inferred latent time equals 0.

Subsequently, for each PCW, we assessed whether macula cells exhibited a higher latent time than peripheral cells for each week. To enhance statistical rigor, all p-values underwent Benjamini-Hochberg correction to obtain adjusted q-values.

### Gene module analysis

To group genes with similar gene expression patterns, gene module analysis was conducted using Hotspot[70]. Hotspot is designed to filter genes that vary significantly and subsequently categorize them into modules.

After latent time was estimated, "informative genes" were characterized as those exhibiting variation with changes in estimated latent time. To achieve this, (1) a similarity map was constructed based on latent time. Subsequently, (2) informative genes were identified by detecting non-random expression patterns within the similarity map, utilizing the "compute_autocorrelations()" function with default parameters. For testing non-random expression patterns, an FDR threshold of 0.05 was employed.

Next, genes were grouped into modules by executing the "create_modules()" function with parameters set to "min_gene_threshold=160" and "core_only=True". By specifying "core_only=True", only genes identified as core genes by Hotspot will be assigned to modules. Genes that not showing any similar patterns to any modules will not be assigned to any modules.

### Gene module score calculation

Additionally, the gene module score was calculated. The gene module score is a per-cell summary score that measures the general pattern of expression for genes in a module. It is the cell-loadings of a single component PCA using smoothed count matrix of all the genes within each module.

### Gene ontology analysis

GO analysis was conducted using g:Profiler[71]. g:Profiler carries out functional enrichment analysis. The analysis was carried out using its online portal (https://biit.cs.ut.ee/gprofiler/gost) with all the default settings, where the input is a list of genes.

### Gene regulatory networks inference

We inferred GRN based on single-cell dual-omic profiles. GRNs were constructed using Pando (v1.0.3)[33]. The process was executed in line with the protocol specified on the Pando website at https://quadbiolab.github.io/Pando/. First, we merged the filtered cells from both RNA-seq and ATAC-seq across all samples separately. For the ATAC-seq integration, we employed Signac (v1.12.0)[72] and adhered to the tutorial provided at https://stuartlab.org/signac/articles/merging/. Specifically, we generated a common peak set using the "reduce()" function from R package GenomicRanges[73] and subsequently merged all cells based on a shared feature set. In addition, peaks that mapped to X and Y chromatin were removed.

Next, the "infer_grn()" function in the Pando package was employed to deduce the GRN. This function was run with its default settings, namely "peak_to_gene_method" set to "Signac" and method configured to "glm", which is a generalized linear model. To refine the GRN edges and find transcription factors (TFs) with high confidence, the "find_modules()" function was employed to filter the network. These parameters used are adjusted p-value cutoff set at 0.1, "nvar_tresh" at 2, "min_genes_per_module" at 1, and "rsq_thresh" at 0.05.

### Gene regulatory networks clustering

After Gene Regulatory Networks were plotted with "plot_network_graph()" function in Pando[33], the clustering of TFs and target genes was performed with K-means algorithm implemented in base R. For K-means algorithm, the number of clusters was identified manually.

### RNA velocity analysis and fate probability inference

To infer cell fate, CellRank V1.5[74] was applied, utilizing a probabilistic approach to assign cell lineage. First, the number of introns and exons was calculated using velocyto[68]. Following this, cell-cell transition probabilities were determined through a combined kernel of (1) "VelocityKernel()" based on RNA velocity (0.8 weighting) and (2) "ConnectivityKernel()" based on similarities among cells (0.2 weighting).

In the analysis of cell transitions among microstates, terminal states were set manually. The "compute_absorption_probabilities()" function with its default parameters was employed to estimate the likelihood of a cell transitioning into various terminal stages. Each lineage fate probability was expressed as a decimal value between 0 and 1, ensuring the cumulative lineage fate probabilities for any given cell always sum to 1.

Finally, specific cell fate probabilities for each cluster were manually determined, allowing the assignment of specific cell fates.

### ATAC−seq trajectory analysis

During ATAC-seq analysis, in order to identify features such as peaks, motif deviations, gene expression, and gene scores that vary from the early to late parts of the trajectory, it is necessary to order the cells. To construct an ATACseq-based trajectory, the "addTrajectory()" function in ArchR[57] was employed. This function involves specifying cells from the earliest time points as the root, and those from the last time points as the terminal. We set cells from PCW 10 as root cells and cells from PCW 23 as terminal cells. Subsequently, ArchR binned the cells based on pseudotime estimated from ATAC-seq UMAP and generated a heatmap displaying the varying features, ordered by pseudotime.

### ATAC motif enrichment and motif deviation analysis

Motif enrichment and deviation analyses were conducted on the pseudobulk peak set. To annotate the peaks, we utilized the Catalog of Inferred Sequence Binding Preferences (CIS-BP) motif from ChromVAR[75]. Furthermore, we computed the chromVAR deviation scores for these motifs using the implementation in ArchR[57]. Subsequently, the motifs were ranked based on the -log10(P-adjusted) Motif Enrichment. To identify genes with highly correlated gene expression and motif deviations, "correlateTrajectories()" in ArchR was used with default parameters.

### Normalization of footprints for Tn5 bias

To remove insertion sequence bias of the Tn5 transposase when estimating TF footprints, the Tn5 bias was subtracts with "plotFootprints()" function in ArchR[57] by specifying "normMethod = "Subtract"".

### Gene regulatory networks visualization

To visualize the inferred gene regulatory network, we used a UMAP embedding to plot the TFs. All the TFs were plotted in a way that reflects the differences in their regulatory effects on target genes. This was achieved using the "get_network_graph()" function in Pando[33].

In the GRN figures, each target gene is colored based on its expression-weighted time, which is a weighted sum of gene expressions relative to sample time. More specifically, for each TF, gene expression was first normalized and scaled to ensure that the sum of gene expressions equals 1. These scaled values were then multiplied by the corresponding sample age (in days) to calculate a gene expression-weighted time.

The size of each TF node represents the average of normalized gene expression values before scaling.

## NRPC TFs cell specification effect prediction and validation

To deduce the effects of TFs on major class specification, we employed the subsequent method: (1) Selection of TF: We first chose all the TFs inferred from the NRPC gene regulatory network (GRN). (2) Average Gene Expression Calculation: Using all annotated NRPC with identified cell fates, we computed the average gene expression for each of these TFs across all cell fate groups. (3) Assignment of TF Specification: A TF was determined to specify a particular major class if it exhibited the highest gene expression value in NRPC destined for that major class.

To corroborate our predictions, we turned to the established literature, specifically examining the documented roles of specific TFs in retina development. (1) Accurate Prediction: If our forecasted major class aligned with those confirmed through loss-of-function experiments in animal models, our prediction was deemed accurate. (2) Incorrect Prediction: If the loss-of-function studies indicated no impact on the major class specification of our predicted major class, our prediction was classified as incorrect. (3) Unknown Prediction: In instances where we could not locate any studies related to a particular TF in the field of retina development, we marked the prediction status as "unknown".

## ATAC−seq peak calling

Initially, peak sets were identified using ArchR[57] through the function "addReproduciblePeakSet()" with all default parameters. This function serves as an R wrapper for executing the MACS2 peak calling pipeline[76]. Throughout this procedure, peaks were called for each major class initially. Subsequently, ArchR assessed the reproducibility of each peak across pseudo-bulk replicates and retained only those peaks surpassing a threshold specified by the reproducibility parameter. Finally, the ultimate set of peaks was identified by combining all peaks from each major class.

## ATAC−seq differential peaks analysis

To perform differential analysis between various major classes and conditions, we utilized the peak set and the getMarkerFeatures() function. Peak intensity was determined as the log2-transformed normalized read counts. Statistical analysis was conducted using the Wilcoxon test and Benjamini-Hochberg multiple test to calculate $p$-values and false discovery rates (FDR) for each pairwise sample comparison. Differentially accessible distal peaks were identified based on predefined thresholds for FDR and log2-fold change for each major class.

## Peak annotation

Peaks were inferred with GREAT[77,78] to predict their biological functions. GREAT was used by providing the peaks as a bed file and the whole genome as a background region through https://great.stanford.edu/great/public/html/.

## Compare ATAC marker peaks with adult

Adult marker peaks were identified using the same differential peak analysis pipeline as the one applied to the adult dataset[2]. Next, for each major class, we checked if a marker identified in the developmental data overlapped with the marker peaks identified in the adult data using the "intersect()" command in bedtools[79]. For each developmental marker peak, if at least 20 percent of the peak region overlapped with any adult marker peaks, it was defined as overlapping; otherwise, it was considered unmapped to adult marker peaks.

## Identification of peak-to-gene linkage

The identification of peak-to-gene linkages entails examining pairs of peaks and genes that display a robust correlation between peak accessibility and gene expression. In this analysis, we established a maximum permissible distance of 250,000 base pairs for each gene while searching for correlated peaks. Additionally, a minimum peak-to-gene Pearson correlation coefficient of 0.45 was set. These criteria were implemented using the "addPeak2GeneLinks()" function in ArchR[57], and the results were subsequently extracted using the "getPeak2GeneLinks()" functions in the same tool.

## Compare Linked Peak's Histone Modification Signals Using Adult Retina Histone Modification Data

H3K27ac and H3K4me2 histone modification regions were downloaded from the previous publication[80]. Next, we checked if a linked peak identified in the developmental data peak-to-gene-linkage analysis overlapped with the peaks identified in the H3K27ac and H3K4me2 region using the "intersect()" command in bedtools[79]. For each developmental-linked peak, if at least 20 percent of the peak region overlapped with any adult retina histone modification peaks, it was defined as overlapping; otherwise, it was considered unmapped.

## Mouse Otx2 cis-regulatory modules to the human genome conversion

13 Mouse *Otx2* cis-regulatory modules were extracted from[45]. Then, those regions were converted from mouse genome mm10 to human genome hg38 using Lift Genome Annotations[81]. All CRMs other than FM2 were successfully mapped to the human genome.

## Enrichment analysis

We performed enrichment analysis as described in ref. 80. 15 eye traits or disorders were from[82–88]. Briefly, we formatted GWAS summary statistics with the MungeSumstats[89], SNPs were then linked to genes using map_snps_to_genes function in MAGAMA[90]. We performed "CellTyping" with default parameters. We formatted snRNA-seq expression data with the EWCE[91], and assessed the linear positive correlation between gene expression major class specificity and gene-level genetic association from GWAS studies by MAGAMA. Celltyping[92] enrichment with FDR < 0.05 were considered as significant.

## Statistics & reproducibility

No statistical method was used to predetermine the sample size. No data were excluded from the analysis. The experiments were not randomized. For all statistical tests used, the data distributions were formally tested and met the assumptions of the statistical tests. Differences were considered significant at $*p \le 0.05$, $**p \le 0.01$, $***p \le 0.001$. Exact $p$-values are indicated in the Source Data file. The investigators were not blinded to allocation during experiments and outcome assessment.

## Reporting summary

Further information on research design is available in the Nature Portfolio Reporting Summary linked to this article.

# Data availability

The RNA-seq and ATAC-seq raw data generated in this study can be accessed at National Center for Biotechnology Information (NCBI) with Sequence Read Archive (SRA) accession ID SRP510712. A copy of raw data has been deposited in the HCA Data Portal - Human Cell Atlas, under accession code 581de139-461f-4875-b408-56453a9082c7 [https://explore.data.humancellatlas.org/projects/581de139-461f-4875-b408-56453a9082c7]. The processed data are available at CZ CELLxGENE Discover with accession code 5900dda8-2dc3-4770-b604084eac1c2c82 [https://cellxgene.cziscience.com/collections/5900dda8-2dc3-4770-b604-084eac1c2c82]. The count matrix for all sequencing data is available at Gene Expression Omnibus, with accession code GSE268630 [https://www.ncbi.nlm.nih.gov/geo/query/acc.cgi?acc=GSE268630]. The binned ATAC-seq peak signaling data

used in this study are available in the UCSC Genome Browser Home under session ID HumanDevelopingRetinaAtlas [https://genome.ucsc.edu/s/zhenzuo2/HumanDevelopingRetinaAtlas]. Adult data used for cell major class annotation and cell subclass annotation is provided in Zenodo [https://doi.org/10.5281/zenodo.10806575]. Adult differentially accessible regions, adult histone modification regions, inferred gene regulatory network model, recovered dynamic models, identified differentially expressed genes model, and bigwig files for ATAC are provided in Zenodo [https://doi.org/10.5281/zenodo.10866348]. Source data are provided with this paper as a Source Data file. Source data are provided with this paper.

## Code availability
Analysis scripts can be accessed with GitHub repository https://doi.org/10.5281/zenodo.11250482[93].

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

## Acknowledgements

This work is part of the Human Cell Atlas (www.humancellatlas.org) and supported by Chan-Zuckerburg Foundation Grant CZF2021-237885 and CZF2019-002425, awarded to R.C. We thank R.C., T.A.R, and A.M. for critically reading the manuscript. Thanks for data curation by Jennifer Zamanian. Thanks for the grammar editing by Aurian Maleki and David Rauch. Figure 1A and Supplementary Figs. 1A, 2A, 3A, 7K, and 9A were created with BioRender.com.

## Author contributions

R.C. conceived and supervised the project. A.M. and A.L. performed the sample collection. Y.L. and X.C. performed the single-nuclei dissociation and sequencing. Z.Z. performed the data analysis. Z.Z., R.C., and S.F. worked on writing, reviewing and editing. T.A.R, J.W., and A.P. shared three early-time datasets. Jin.L. provided analysis support and made data available online. J.S. performed GWAS enrichment analysis. Y.B. and Jean.L. prepared Supplementary Data 8. Jiaxiong.L. confirmed the accuracy of Supplementary Data 8. M.T. provided Macular Hypoplasia Related Genes. All authors approved the manuscript. All authors edited the manuscript.

## Competing interests

The authors declare no competing interests.
