## [Peer Review File · Nature Communications]

REVIEWER COMMENTS

Reviewer #1 (Remarks to the Author):

This study presents a detailed single-nuclei multiome atlas of the human developing retina, covering various stages of development and different regions (macular and peripheral) of the retina. This comprehensive atlas is a significant achievement in itself.

It in fact provides a wealth of information on the gene expression profiles and open chromatin regions of various cell types within the developing retina, offering a deep dive into retinal development.

The work also highlights the temporal and spatial dynamics of different cell types within the retina. The identification of distinct gene modules and their associated biological processes in photoreceptor progenitor cells (PRPC) is a remarkable finding not only because it sheds light on the regulatory networks governing retinal development but also because can be used as a bioinformatics example of how the use of gene modules help these types of analyses.

The analysis of cis-regulatory regions and their overlap with adult retinal regions, as well as histone modification patterns, is another important aspect of this study. It connects epigenome regulation to gene expression during development.

The data provided by this work has the potential to significantly contribute to the field of retinal development and related areas. The comprehensive single-nuclei multiome atlas of the human developing retina is a valuable resource because it can serve as a reference point for future studies.

The temporal and spatial data can be used for understanding how retinal diseases and disorders develop and potentially lead to the development of therapeutic interventions.

I really liked the way the analysis was set up for the identification of gene modules and regulatory networks because this can be applied for other type of similar analyses and so advance the field not only for the results (that help the molecular mechanisms governing cell fate determination) but also from a methodology perspective.

I believe the work is original and takes into account the literature.

The paper relies heavily on computational and analytical methods. More experimental validation, such as functional assays, perturbation studies, or in vivo experiments, would definitely improve the robustness of the findings. Additionally, the paper is also missing some in-depth functional interpretation of these findings.

While the work is significant and promising, it's important for the research community to continue building upon these findings. Future studies can delve deeper into the functional roles of specific genes and regulatory networks identified in this work and further explore the implications for retinal diseases and regenerative therapies.

By looking at the scripts, workflow and how data are presented, I could not find any flaw in the data analysis. The github page and the data provided allow the reproducibility of the study which is great.

Reviewer #2 (Remarks to the Author):

This is an elegant study that generated an important high-quality dataset to profile human developing retina at single cell resolution. The study provided important insights into the epigenetic and transcriptomic landscape of human fetal retinal development and the different molecular mechanisms and kinetics in driving macula and peripheral retina formation. The manuscript is well written and the study design is logical. The conclusions are mostly supported by solid data. The curated dataset would serve as an important resource to study retinal development for many researchers in the field and is certainly a welcome addition to the literature. However, there are a few comments that the authors should address to improve the readability of the manuscript. The specific comments are:

1. I would suggest to avoid overstatement by reporting the post-QC nuclei number for the final dataset (~205k) in the abstract, summary + Fig1A, rather than the pre-QC nuclei number (300k).
2. Can the authors comment on why a large number of nuclei (~30% of processed nuclei) didn't pass QC? What are the potential reasons causing this?
3. Y axis label for Fig 1C-D => majorclass should be two words
4. Fig 2K: improve figure presentation for (1) and (2). I.e scale for % cells in (2) is not readable.
5. Fig 3F: % cell scale for 3F is not readable

6. Fig 3H: I would encourage the author to elaborate on the discussion for the identified GRNs in NRPC with different retinal cell fates, especially the misassigned TFs in red. For instance, in 'Rod' there are more misassigned TFs (red) than correctly assigned ones (green), and > 50% of the TFs are novel with unknown association to rod, which impacts on the confidence that the annotation/clustering for NRPC with different retinal cell fates are done accurately.
7. Fig 5: Can the author explain the significance of the PRPC genes which are mostly primed and coupled-on? Are these genes similar to poised/bivalent genes in embryonic stem cells that can be rapidly activated in response to differentiation?
8. Line 318: 'GRID2 displays notably high expression exclusively in MGs within the macular region'. This statement implies that GRID2 is expressed exclusively in macular MGs, however the result showed that GRID2 is expressed in both peripheral + macular MGs (Fig 6G).
9. Fig 7A: In addition to showing the expression of foveal hypoplasia genes in major retinal cell types, which presumably one can extract with other adult/fetal retina gene datasets too, can the authors look at the expression of selected foveal hypoplasia genes at different PCW timepoints? This might provide some interesting info to when foveal hypoplasia might manifest during development.
10. Did the author look at RPE in the developing retina dataset?

Reviewer #3 (Remarks to the Author):

This manuscript examines single-nucleus transcriptome and chromatin accessibility profile of developing human retinae, utilizes the generated "multi-omics atlas" to study cell fate trajectories across developmental stages and locations in the retina, and further explores the gene regulatory networks that drive cell fate transitions. Overall, this study provides a valuable resource for understanding of retinal development and related diseases and will be of importance. The efforts invested in compiling and generating this data are commendable, and the information produced should be of value. However, many of the conclusions are not justified by the new data being presented. The manuscript is also rather complex in organization and requires careful editing. Improvements are needed in clearly delineating novel contributions, biological validation of new results, and a more in-depth analysis that justifies tool selection and integrates results into a cohesive story. Additionally, the figures are not sequentially arranged, and some of the supplementary figures are not even referenced in the main text.

Major Critiques:

Title: The title appears profound and pretentious. The word “Multiome” is inaccurate though many seem to be using it since only two omics datasets (transcriptome and ATAC-seq) are used. Same is true for “Atlas” since only 12 donor retinas from 12-23 weeks post-conception were used in the study. To use “Multiome” and “Atlas,” one must have additional omics datasets and many more samples. The data being presented is significant and useful but limited.

Conclusions and novelty. For the most part, the authors are describing some primary results obtained from the data. There is a lack of in-depth exploration, and the authors' presentation of these results lack an intrinsic logical connection, making it difficult to draw definitive conclusions. For example, the comparative analysis of RNA-seq data from the macular and the peripheral retina in developing human eye has been analyzed previously by other groups. In addition, the involvement of the retinoic acid pathway is not a new or innovative finding, but rather supports published findings. Thus, it is crucial to contrast the data and results of the manuscript with a considerable amount of pre-existing published data, some of which was reported by the authors' laboratory. The main claim of the current manuscript appears to be the addition of snATAC-seq data. However, snATAC-seq alone, without integration of additional 'omics' data, does not greatly add to new mechanistic insights. While the authors occasionally allude to the consistency of their results with previous knowledge, they should explicitly articulate the new insights being reported in this manuscript by addition of new data.

Power: The authors utilize a relatively small number (12) of donors. The human donors (unlike laboratory models) have considerable variance to mask or exaggerate the differential outcome observed by the analysis being employed in the manuscript. The authors should either establish that the inter-donor variability is insignificant or must employ methods to mitigate such variations. Or, additional samples are needed.

Rationale for using different analytical methods: The manuscript falls short in delivering a cohesive and comprehensive analysis of the data. The impression is given that the authors applied tools available in the literature without adequately justifying their choices or providing a detailed account of the rationale behind selecting specific tools over alternatives. A more thorough discussion is needed, addressing why certain tools were chosen, the significance of their application, and how the results from various tools contribute to a comprehensive and coherent narrative.

RNA velocity analysis. The authors employed RNA velocity for trajectory inference analyses on multiple occasions. RNA velocity is based on measurements of spliced and unspliced mRNA transcripts to predict the direction of future gene expression changes in the cell. In scRNA-seq, RNA is collected from the whole cell, whereas in snRNA-seq, it is collected from the nucleus. This means that snRNA-seq may capture less cytoplasmic RNA and more unspliced nascent RNA, additionally the RNA degradation rate therein may be replaced by the rate of RNA exporting the nucleus. There is concern about the applicability of this model to the snRNA-seq data. For example, in Figure 2B, the direction indicated by arrows in the upper left part of PRPC seems to be opposite

to the real differentiation direction of PRPC in the Figure 2A. The authors need to make necessary adjustments to ensure the accuracy of the results. Did the authors perform multiple trajectory inference methods to analyze and validate their results? In addition, the authors have real chronological data of developing retina; could they indicate to what extent do their predicted directions of cellular differentiation correspond to the real direction of retinal development?

Submission of raw data. In the Data Availability part, the authors claimed that all raw data and processed data can be accessed in HCA Data Portal. But this reviewer can only see part of the processed data from that link rather than all the raw data. It only contains 205,619 cells after quality control instead of the original 296,158 nuclei they profiled. The authors are requested to submit the necessary raw data. In addition, the authors mentioned that they employed in house adult human retina derived data to perform the Major Class Annotation and Maturation Score Calculation. These are key factors that relate to the conclusions of the manuscript and therefore it is necessary for the authors to submit and clarify the raw data of adult cells they used.

Experimental validation: The study mentions many transcription factors and gene regulatory networks; however, most of these findings are inferred or previously predicted. Some experimental validation of new findings is essential for linking them to relevant biological context. The results of the analysis, particularly those included in the extensive lists in several supplementary tables, lack biological validation.

Limitations: The authors should provide a more balanced view by discussing more explicitly the limitations of their methodology and findings.

Additional specific comments:

1. To understand the analysis better, the authors should consider providing a workflow diagram.
2. The manuscript mentions that a total of 51 cell types and 5 precursor groups were identified. How does this list compare with the comprehensive list of cell types curated by Sanes Lab and others in different organisms [Table 1 in <https://www.annualreviews.org/doi/full/10.1146/annurev-vision-032621-075200>]. That table does not include cell types from developing retina and should be extended.
3. The manuscript also mentions that these cell types are hierarchical. It is imperative that the authors include a tree diagram of the cell types with chronological information as provided in Figure 4c or Figure 6g in Qui et al preprint from Shendure lab (<https://www.biorxiv.org/content/10.1101/2023.04.05.535726v1.full>)

4. While annotating the cells into several major and sub cell types, it seems that the authors have utilized several tools and applied manual curation. However, the exact steps are unclear from the information provided in the main text and the supplement. To give an example, when the authors subdivided the Retinal Progenitor Cells (RPC) into Primary RPC (PRPC), Neuronic RPC (NRPC) and Mueller Gila (MG) how were the boundaries decided. I guess that some scaffolding was done by clustering those cells and the clusters were annotated with one of these sub cell types, but that crucial information is missing from the description.

5. The authors have used several pieces of information such as pseudo time, latent time and emission probability derived from tools such as multivelo, scvelo, monocle and cell rank. However, it is not clear what exact information from each of these tools were used. It is not clear if only one of the tools such as CellRank or scVelo would have been sufficient (e.g. CellRank version 2 internally derives RNA velocity and similar information). Moreover, some of the tools have been mentioned in multiple sections in the materials and methods section making it more confusing why and in which order the tools were used. For example, MultiVelo has been mentioned in the subsection “Latent Time Inference”, “Multi-omic Dynamical Modeling” and “RNA Velocity Analysis and NRPC Fate Inference”. Why? My guess is that the tool was applied only once, and the results were reused multiple times. The authors should clarify that or simplify the methods sections by proper reorganization. Similar can be said about using monocle3 in “Maturation Score Calculation” and “Pseudotime Analysis”.

6. When the authors use a particular tool or does some analysis, the authors should make that analysis as much comprehensive as possible. For example, in the results section “AC Subclasses Formed from Specific Groups of NRPC at Different Times” the authors mention how a subgroup of Neurogenic RPCs differentiate to different Amacrine subtypes such as Starburst, GABAergic, Glycinergic and Dual ACs. But I guess that similar analysis can be done for NRPCs that differentiate to other major cell types such as photoreceptors, bipolar cells, and so on. Why was not similar analysis done?

7. The authors should provide a smooth transition when changing from one result section to another.

8. In all vertebrates, retinal cells are demonstrated to be produced in a conserved order with RGCs being differentiated first and Muller Glia at a late stage. Is it correct that, in Figure 2, the authors imply that MG are differentiated first? This needs a careful analysis and discussion. Again, validations are required for any such conclusions about trajectory of cell birth.

Minor comments:

1. The word “region” seems to have a dual meaning in this manuscript: 1) open chromatin region, and 2) region within the retina – macula or periphery. It would be good to use “spatial location” or something similar for the second case.

2. The words reference/query and adult/fetal has been used interchangeably in the manuscript. It will be good to follow a consistent nomenclature; preferably adult/fetal.
3. How were gene activity scores in the UMAP of Fig. 1D generated?
4. When referencing first time in line 94 on page 3, please cite the reference atlas.
5. Typo error: In Figure 2C, “Latet” should be “Latent”. Supplementary Fig.1F should be in 1E.
6. In Figures 2C and 2D, there is a lack of necessary introduction about the latent time. In other words, what does the value of the latent time mean? In addition, the conclusion for Figure 2D (Page 4 Line 122-125) lacks the necessary statistical description and is not rigorous enough.
7. Page 4 Line 135: The authors indicate Supplementary Fig.4A when they describe Module 2, but the Supplementary Fig.4A lacks necessary descriptions. It is confusing. I don't understand what this indicator is meant to illustrate.
8. Supplementary Fig.4B-C: What does the value of the y-axis mean? In addition, it is necessary for the authors to describe in detail how they got the conclusions in Page 4 Line 141-143.
9. The description of some figure legends are too simple and lacks necessary explanations. For instance, lack of legend for colors: Figure 2I, Figure 3E, supplementary Fig.5C, Figure 4H.
10. Page 5 Line 174 : The authors showed identified differentially expressed genes of different cell cluster in Fig.3F and Supplementary Fig.5C, which could be interpreted as potential genes to drive neural type specification, however it does not seem to involve pathways. There is a suspicion of over-interpretation here.
11. Figure 5D: Where did the histone modification data originate? Also please provide a description of the analysis process for this figure in the Methods section.
12. Figure 5G: It looks confusing and not well described in the legend. The gray highlighted area indicates a cone-specific OCR region, yet this region doesn't seem to be a cis-regulatory module because it doesn't have a peak. Also, what do the asterisked peaks mean? The description for this figure is at Page 7 Line 263-268, but I'm not sure what the authors are trying to explain. The authors showed an example of OCRs near the OTX2 gene loci. Are they trying to show that these OCRs are associated with differential expression in Figure 5F? If so, they should provide the appropriate argumentation process.
13. Page 8 Line 290 / Fig.6A and Supplementary Fig.9 and 10: The examines were performed in all cell types rather than only PRPC according to the figures. Page 9, line 329 and Figure 7A – AP3B1 seems to be high in all cell classes.
14. Page 14, line 523 – enlist and cite cell cycle genes
15. Page 14, line 525 – Six PCW should be five PCW
16. Page 14, line 556 – cite “established marker genes”
17. Page 15, line 565 – percentile better than quantile

18. Page 15, line 575 – I do not understand why we need to fit, as mean and standard deviation completely determines the density. Right?
19. Page 15, line 592-595 – It would be good to show monocle pseudotime in a plot somewhere.
20. Page 18, line 679 – “g Profiler” -> “g:Profiler”
21. Page 20, line 766 – Fig. 1H should be Fig. 1C
22. Page 21, lines 792-793 – “UMAPs on the right represent cells with highest module scores for each of the identified modules” -> “UMAPs on the right show a cell with the color of the module among all identified modules that has highest score for that cell” or something like that. Also, these UMAPs are confusing. I would suggest including all cells but showing irrelevant cells in gray color. BTW, the methods section should explicitly mention how these module scores are computed.
23. Page 22, line 866 – how are motif deviations computed?
24. Page 23, line 884 – Log2 of fold change?
25. Page 23, line 889 – What does “red” group of DARs signify?
26. Page 23, line 892 – How do you obtain functions of cis-regulatory regions? I guess it is different than GO terms mentioned in methods section.
27. Page 23, line 898 – Please cite from where you obtained histone modification information.
28. Page 23, line 900 – Provide more information about linked peak – how are they computed?
29. Page 23, line 898 – Please cite from where you obtained cis-regulatory information.
30. Page 23, line 918 – Explain “priming in total gene time”
31. Fig 1c/d – Consider using better color scheme
32. Fig 1H – Consider using continuous time scale instead of discrete. Preferably using number of days past conception.
33. Fig 2E – Why is the blue annotation bar on the left of heatmap not extending all the way to the bottom? Are there genes outside the three modules?
34. Fig 2G – provide gene expression color legend
35. Fig 2H – why is not the GRN showing edges between genes?
36. Fig 3E – provide gene expression /emission probability color legends
37. Fig 3G – why is not the GRN showing edges between genes?
38. Fig 3J – Why do the UMAPs look different than 3A?
39. Fig 4B – Can the two plots have same Y-axis scale for better comparison of the relative value of density?
40. Fig 4D – Are the UMAP coordinates of these two plots different from panel A?

41. Fig 4H – Are all the points in a single UMAP or the three modules have separate UMAPs?
42. Fig 5 – It is well known that as the cells differentiate, the number of open chromatin peaks are reduced. There should be plot to show the same in this data.
43. Fig 6A – which sub panel is for macula, and which is for periphery?
44. Supp Fig 1A – the figure legend and text have reversed meaning of the colors.
45. Supp Fig 1C/D – the UMAP does not look like the bown part of UMAP in SF1B
46. Supp Fig 1F – F missing on figure.
47. Ensure color legends in all figures.

Reviewer #4 (Remarks to the Author):

I co-reviewed this manuscript with one of the reviewers who provided the listed reports as part of the Nature Communications initiative to facilitate training in peer review and appropriate recognition for co-reviewers.

Reviewer #5 (Remarks to the Author):

I co-reviewed this manuscript with one of the reviewers who provided the listed reports as part of the Nature Communications initiative to facilitate training in peer review and appropriate recognition for co-reviewers.

Reviewer #1 (Remarks to the Author):

This study presents a detailed single-nuclei multiome atlas of the human developing retina, covering various stages of development and different regions (macular and peripheral) of the retina. This comprehensive atlas is a significant achievement in itself.

It in fact provides a wealth of information on the gene expression profiles and open chromatin regions of various cell types within the developing retina, offering a deep dive into retinal development.

The work also highlights the temporal and spatial dynamics of different cell types within the retina. The identification of distinct gene modules and their associated biological processes in photoreceptor progenitor cells (PRPC) is a remarkable finding not only because it sheds light on the regulatory networks governing retinal development but also because can be used as a bioinformatics example of how the use of gene modules help these types of analyses.

The analysis of cis-regulatory regions and their overlap with adult retinal regions, as well as histone modification patterns, is another important aspect of this study. It connects epigenome regulation to gene expression during development.

The data provided by this work has the potential to significantly contribute to the field of retinal development and related areas. The comprehensive single-nuclei multiome atlas of the human developing retina is a valuable resource because it can serve as a reference point for future studies.

The temporal and spatial data can be used for understanding how retinal diseases and disorders develop and potentially lead to the development of therapeutic interventions. I really liked the way the analysis was set up for the identification of gene modules and regulatory networks because this can be applied for other type of similar analyses and so advance the field not only for the results (that help the molecular mechanisms governing cell fate determination) but also from a methodology perspective.

I believe the work is original and takes into account the literature.

The paper relies heavily on computational and analytical methods. More experimental validation, such as functional assays, perturbation studies, or in vivo experiments, would definitely improve the robustness of the findings. Additionally, the paper is also missing some in-depth functional interpretation of these findings.

While the work is significant and promising, it's important for the research community to continue building upon these findings. Future studies can delve deeper into the functional roles of specific genes and regulatory networks identified in this work and further explore the implications for retinal diseases and regenerative therapies.

By looking at the scripts, workflow and how data are presented, I could not find any flaw in the data analysis. The github page and the data provided allow the reproducibility of the study which is great.

We thank the reviewer for the positive comment for our work. We have also added additional experimental data for validation.

1. During this revision, three additional multi-omics datasets from two donors were integrated into our original dataset generated by our lab for this study. These additional datasets were generated and shared by a different laboratory (Tom Reh lab) using the same technologies but with totally different experimental equipment, sequencing machine, and technicians. Despite these variations, the seamless integration of the newly added datasets with our original dataset serves as validation for the robustness of our data.

Three UMAPs, as depicted in the following figure, serve as evidence of the effectiveness of our data integration performance.

Here are three UMAPs showing the integrated data from our lab and data from Tom Reh Lab. The first UMAP was colored by Lab ID, distinguishing between Rui Chen Lab (in blue) and Tom A Reh Lab (in orange). The second UMAP was colored by Sample ID, and the third UMAP was colored by sample age. This integration suggests that our data is not influenced by technical variations, individual differences, or batch effects. Instead, the UMAP is primarily driven by cell types and ages, thereby further enhancing the reliability of our results.

2. In the previous section, we have shown the trajectory of our data on the UMAP is primarily influenced by the temporal dynamics of developmental gene expression rather than batch effects. To further prove this observation, instead of using UMAP, we utilized adult data to compute the maturation score—a measure of gene expression similarity between adult and developmental data. Across all major classes, the cell maturation score consistently increases over time, signifying a progression toward greater cellular maturity from samples collected at an early age to those obtained later in life, as

expected. This observation is further supported by the expression of marker genes associated with maturation, which have been documented in previous literature.

- Next, we would like to compare the two modalities observed in our multiome data. By graphing the top marker genes identified for each major class using both RNA-seq gene expression and plotting the corresponding ATAC-seq gene score of those marker genes, we observed consistent patterns between the two modalities. This validation indicates that the two modalities agree with each other, showing the overall high quality of the data.

Those strong evidence can validate the quality of this dataset.

Reviewer #2 (Remarks to the Author):

This is an elegant study that generated an important high-quality dataset to profile human developing retina at single cell resolution. The study provided important insights into the epigenetic and transcriptomic landscape of human fetal retinal development and the different molecular mechanisms and kinetics in driving macula and peripheral retina formation. The manuscript is well written and the study design is logical. The conclusions are mostly supported by solid data. The curated dataset would serve as an important resource to study retinal development for many researchers in the field and is certainly a welcome addition to the literature.

We thank the reviewer for the positive comment.

However, there are a few comments that the authors should address to improve the readability of the manuscript. The specific comments are:

1. I would suggest to avoid overstatement by reporting the post-QC nuclei number for the final dataset (~205k) in the abstract, summary + Fig1A, rather than the pre-QC nuclei number (300k).

We appreciate the valuable feedback. In response to their comment, we have updated the manuscript by replacing the initial count of nuclei before quality control (QC) with the count of nuclei after QC, as recommended. This adjustment has been made consistently throughout the abstract, main body of the article, supplementary materials, and Figure 1A. Please refer to the revised Figure 1A presented here, reflecting the implemented changes.

	Location	Method	Number of Nuclei After QC
PCW8	Whole Eye	Sn-Multiomics	6,863
PCW10	Macula and Periphery	Sn-Multiomics	43,056
PCW13	Macula and Periphery	Sn-Multiomics	40,976
PCW15	Macula and Periphery	Sn-Multiomics	41,025
PCW19	Macula and Periphery	Sn-Multiomics	60,617
PCW23	Macula and Periphery	Sn-Multiomics	33,969

2. Can the authors comment on why a large number of nuclei (~30% of processed nuclei) didn't pass QC? What are the potential reasons causing this?

We appreciate the valuable feedback. The considerable number of nuclei did not meet quality control (QC) standards, primarily due to the dual modalities in our dataset. Given the absence of a universally accepted pipeline for multi-omics data QC, we implemented a rigorous pipeline to filter nuclei based on BOTH gene expression and chromatin state. Only nuclei that successfully passed through all flowing filtering steps were kept for subsequent analyses.

In detail, our data underwent four filtering processes, ensuring the robustness and reliability of the retained dataset. In each filtering step, only nuclei that passed the previous filtering were considered for further filtering.

1. Filter 1: Filter based on Gene Expression Data
For each cell, genes numbering less than (or equal to) 500 or exceeding (or equal to) 10,000 were removed. Additionally, cells with greater than 10% mitochondrial counts were filtered out. This step utilized the Seurat function 'subset()' with parameters as following $nFeature_RNA > 500 \ \& \ nFeature_RNA < 10000 \ \& \ percent.mt < 10$.
4. Filter 2: Filter doubles with DoubletFinder V2.0 on RNA-seq
Doublet identification was performed based on gene expression data using parameters determined through pN-pK parameter sweeps and mean-variance-normalized bimodality coefficient (BC_{mvn}). Furthermore, during the parameter selection process, a maximum of 7.5% of doublets were allowed to prevent excessive data loss.
5. Filter 3: Doublet Inference with ArchR on ATAC-seq
Utilizing ATAC-seq data, doublet scores were estimated with default parameters in ArchR.

6. Filter 4: Annotation-Based Nuclei Removal

Nuclei annotated as retinal pigment epithelium and microglias were excluded from the analysis.

The table below summarizes the number of nuclei retained after each filtering step for each sample:

Sample ID	Donor ID	Time	Region	Days	Number of Nuclei Sequenced	Pass Filter 1	Pass Filter 2	Pass Filter 3	Pass Filter 4
sn_multitome_d59		1 8w3d	Whole Eye	59	8927	8674	8645	7128	6863
Multitome_10w_FR		2 10w	Macula	70	6189	6093	6047	5214	4972
Multitome_10w_NR		2 10w	Peripheral	70	4717	4467	4151	3764	3578
sn_multitome_d76c		3 10w6d	Macula	76	5252	4506	4489	3914	3814
sn_multitome_d76p		3 10w6d	Peripheral	76	4840	4514	4499	3502	1400
Multi_Fetal_11w2d_FR		4 11w2d	Macula	79	12692	12179	11161	9232	8916
Multi_Fetal_11w2d_FR_2		4 11w2d	Macula	79	15964	15277	14006	11522	11031
Multi_Fetal_11w2d_NR		4 11w2d	Peripheral	79	12675	12455	12410	10039	9345
Multitome_12w3d_FR		5 12w3d	Macula	87	8522	8400	7673	6621	6382
Multitome_12w3d_NR		5 12w3d	Peripheral	87	9318	8681	8121	6673	6134
Multi_Fetal_13w_FR		6 13w	Macula	91	17628	17426	15946	12961	12642
Multi_Fetal_13w_NR		6 13w	Peripheral	91	20000	19876	19795	16727	15818
Multitome_14w2d_FR		7 14w2d	Macula	100	6300	6132	5635	4931	4733
Multitome_14w2d_NR		7 14w2d	Peripheral	100	7414	7245	7217	6240	5942
Multi_Fetal_14w5d_FR		8 14w5d	Macula	103	20000	19383	19220	15621	14771
Multi_Fetal_14w5d_NR		8 14w5d	Peripheral	103	9195	9023	8987	7546	7100
Multitome_16w4d_FR		9 16w4d	Macula	116	5932	5773	5747	4990	4310
Multitome_16w4d_NR		9 16w4d	Peripheral	116	5375	5243	5223	4358	4169
Multi_Fetal_19w4d_FR		10 19w4d	Macula	137	20000	19163	19078	15623	14257
Multi_Fetal_19w4d_NR		10 19w4d	Peripheral	137	19190	18927	17401	14317	14142
Multitome_20w1d_FR		11 20w1d	Macula	141	6466	6265	6238	5536	4879
Multitome_20w1d_NR		11 20w1d	Peripheral	141	7529	7359	6840	5593	5490
Multi_Fetal_20w2d_FR		12 20w2d	Macula	142	14273	13973	13722	10748	9475
Multi_Fetal_20w2d_NR		12 20w2d	Peripheral	142	17708	17287	16133	12655	12374
Multi_Fetal_23w1d_FR		13 23w1d	Macula	162	20000	19333	19249	15638	14473
Multi_Fetal_23w1d_NR		13 23w1d	Peripheral	162	15725	15340	14071	10778	10484
Multitome_23w4d_FR		14 23w4d	Macula	165	6461	6189	6134	4913	4456
Multitome_23w4d_NR		14 23w4d	Peripheral	165	6885	6322	6296	5258	4556
			Total		315177			Total	226506

You may notice within this process we performed doublet removal twice to ensure the effective elimination of doublets. We did first doublet removal on RNA-seq and second doublet removal on ATAC-seq. We did this because doublets pose a significant challenge in developmental studies, as distinguishing between doublets and those in transition to a differentiated stage can be complex.

This table is also available at Supplementary Table 2.

3. Y axis label for Fig 1C-D => majorclass should be two words

We appreciate the reviewer's observation and acknowledge the oversight. In response, we have revised the figure to enhance clarity by removing row labels. This adjustment aims to streamline the presentation and maintain a clean visual representation. Here is the panel before revision. The color code for the major class is not consistent with the previous panel.

In this updated figure, we have omitted the y-axis label to enhance the overall cleanliness of the visualization. Additionally, compared to the figures prior to the revision,

we have implemented imputation on gene scores/gene expression, resulting in an improved and clearer representation.

In the meanwhile, “Majorclass” has been corrected to “major class” for the rest of the manuscript.

4. Fig 2K: improve figure presentation for (1) and (2). The scale for % cells in (2) is not readable. We thank the review for pointing it out and we improved the figure accordingly.

The above panel displays the panel prior to revision. It appears to be confusing due to the absence of legends for certain subpanels and the insufficient size of figure text for some panel titles.

The panel above is the revised one. Now, the two panels have been merged into a singular display. Each subpanel is accompanied by a legend featuring a color scale. The primary emphasis of the first subpanel lies in showing the probability of fate into NRPCs (from PRPCs), while the subsequent subpanels show marker genes' expression. Those

marker genes are chosen because they are highly correlated with inferred fate probabilities. It is noteworthy that all subpanels employ the same UMAP for PRPCs, contributing to an improved understanding of the plotted UMAP for the reader.

5. Fig 3F: % cell scale for 3F is not readable

We value the feedback from the reviewer and have implemented improvements to the figure, as depicted in the following illustration. In the initial version (displayed on the left), panels exhibited variations in size, causing difficulties in reading certain text. In the revised version (shown on the right), we have reduced the legend text size, ensuring that all text elements do not overlap. As a result, readers can now experience a more visually effective presentation.

6. Fig 3H: I would encourage the author to elaborate on the discussion for the identified GRNs in NRPC with different retinal cell fates, especially the misassigned TFs in red. For instance, in 'Rod' there are more misassigned TFs (red) than correctly assigned ones (green), and > 50% of the TFs are novel with unknown association to rod, which impacts on the confidence that the annotation/clustering for NRPC with different retinal cell fates are done accurately.

We thank the reviewer for providing valuable comments. In summary, our analysis indicates that 22 out of 32 transcription factors (TFs) were accurately predicted at the major-class-level fate specification function during development, while 10 out of the 32 TFs were mis-predicted. We acknowledge several potential factors contributing to these mispredictions:

1. Complexity of Gene Regulation: The process of gene regulation during development is redundant and dynamic. It is possible that our current TF function inference method may be too simple to effectively handle such a complex system.
2. Inference Method for Negative Regulations: Our TF function inference relied on TF expression levels within each NRPC group. The prediction was based on the group with the highest average expression level, assuming positive regulation. This hypothesis may overlook negative regulations, leading to mispredictions.
3. Temporal Limitations: Our current TF function inference tool primarily accounts for early regulations occurring in NRPCs. If cell specification occurs at later time points or if fate shifting happens during differentiation, our model may not accurately predict these events.

4. **Dependency Limitations:** Our existing TF inference tool operates on a single TF at a time, overlooking the collaborative nature in which TFs often function as a team with other co-regulators to achieve specific cellular functions.

Addressing the gene regulation dynamics during development is a complex task and extends beyond the scope of the current paper. Future research is needed to have a tool developed for more comprehensively capturing the dynamic developmental processes.

7. **Fig 5:** Can the author explain the significance of the PRPC genes which are mostly primed and coupled-on? Are these genes similar to poised/bivalent genes in embryonic stem cells that can be rapidly activated in response to differentiation?

Thank you for bringing this matter to our attention. In our investigation of marker genes expressed during embryonic stem cell development within the context of retina development, we observed that certain genes, such as *MECOM*, were only minimally expressed in our datasets, as illustrated in the UMAP visualization below. This indicated our data may not capture the whole expression cycle of those genes. Also, due to the limited presence of spliced and unspliced reads associated with this gene, we were unable to recover its dynamics accurately.

Another challenge we have is not all genes can be fitted to the dynamic modeling. As a total of around 2,200 temporal-dynamics genes, primed stages were detected in 645 genes.

For significance, we compared the gene expression profiles of early progenitors to those of late progenitors. Interestingly, we discovered that genes expressed in early progenitors exhibited shorter durations of coupled-on stage compared to those expressed in late progenitors. Our analysis suggests that gene expression patterns align more closely with chromatin accessibility in later stages rather than in earlier stages, a trend that aligns with expectations.

8. **Line 318:** 'GRID2 displays notably high expression exclusively in MGs within the macular region'. This statement implies that GRID2 is expressed exclusively in macular MGs, however the result showed that GRID2 is expressed in both peripheral + macular MGs (Fig 6G).

Thank you for bringing this to our attention. In response, we have clarified the GRID2 as a DEG, specifying GRID2 is a DEG within MGs and not within any other cell types. So,

you may notice GRID2 express in many other cell types. To avoid potential confusion, we have omitted this information in the revised version. Instead, we have incorporated a

heatmap illustrating the DEG status of genes across major classes. This heatmap allows readers to discern the overall distribution of DEGs within each major class, highlighting exclusive DEGs for each class and those shared among different major classes.

9. Fig 7A: In addition to showing the expression of foveal hypoplasia genes in major retinal cell types, which presumably one can extract with other adult/fetal retina gene datasets too, can the authors look at the expression of selected foveal hypoplasia genes at different PCW timepoints? This might provide some interesting info to when foveal hypoplasia might manifest during development.

Thank you sincerely for your insightful suggestions. Our investigation focused on the expression of two foveal hypoplasia genes within the retina, aiming to discern any temporal patterns throughout development. Remarkably, we found CNGB3 and PDE6H have dynamic gene expression patterns in cones. They not only exhibit spatial differences—revealing significantly higher expression in macula samples compared to peripheral ones—but also start to express highly after PCW 13 in the macula and after PCW 15 in the periphery.

We also repeated the same analysis on RA pathway-related genes in Fig 6 and found

interesting patterns. TBX5 and ALDH1A1 showed positively correlated gene expression patterns, consistent with previous research.

Spatiotemporal Gene Expression in PRPCs

10. Did the author look at RPE in the developing retina dataset?

Thank you for highlighting this aspect. Due to the limited numbers of RPEs in our current dataset, we did not specifically investigate RPE development in this study. Additionally, the selected time frame for sample collection (PCW 8 to PCW 23) was optimized for capturing retina development, potentially missing the early stages of RPE development (PCW5 to PCW 8), as RPEs tend to develop earlier than the retina. Future studies with a dedicated focus on RPE development and an adjusted sampling strategy may provide more comprehensive insights into the dynamics of retinal and RPE development. We have recently generated a dataset specifically focusing on development RPE and are working on the data analysis. We hope to share our findings as soon as we can.

Reviewer #3 (Remarks to the Author):

This manuscript examines single-nucleus transcriptome and chromatin accessibility profile of developing human retinae, utilizes the generated “multi-omics atlas” to study cell fate trajectories across developmental stages and locations in the retina, and further explores the gene regulatory networks that drive cell fate transitions. Overall, this study provides a valuable resource for understanding of retinal development and related diseases and will be of importance. The efforts invested in compiling and generating this data are commendable, and the information produced should be of value.

We thank the reviewer for the positive comment.

However, many of the conclusions are not justified by the new data being presented. The manuscript is also rather complex in organization and requires careful editing. Improvements are needed in clearly delineating novel contributions, biological validation of new results, and a more in-depth analysis that justifies tool selection and integrates results into a cohesive story. Additionally, the figures are not sequentially arranged, and some of the supplementary figures are not even referenced in the main text.

Major Critiques:

Title: The title appears profound and pretentious. The word "Multiome" is inaccurate though many seem to be using it since only two omics datasets (transcriptome and ATAC-seq) are used. Same is true for "Atlas" since only 12 donor retinas from 12-23 weeks post-conception were used in the study. To use "Multiome" and "Atlas," one must have additional omics datasets and many more samples. The data being presented is significant and useful but limited.

We appreciate your valuable suggestions. In response to your insights, we have incorporated three additional samples from two donors to extend the dataset's temporal coverage. Originally spanning from post-conception week 10 to 23, the dataset now includes an additional sample from post-conception week 8 and two samples from PCW 10.

The term "Multiome" used in our study originates from the 10X product (<https://www.10xgenomics.com/products/single-cell-multiome-atac-plus-gene-expression>), which we have acquired. It's important to note that we did not coin this term ourselves. "Multiome" has been widely used in most recently published studies (DOI: 10.1126/sciadv.adg3754, 10.1038/s41586-022-05279-8), so we followed this naming. In this context, "Multi" signifies the incorporation of more than one modality, marking a substantial advancement over traditional single modality approaches.

In the meanwhile, we like to address that obtaining human developmental samples is a great challenge, particularly when aiming to cover an extensive time span covering all key developmental events. Hence, we refer to our dataset as an "atlas" due to its size, exceedingly even the largest datasets previously published (more than doubled).

Conclusions and novelty. For the most part, the authors are describing some primary results obtained from the data. There is a lack of in-depth exploration, and the authors' presentation of these results lack an intrinsic logical connection, making it difficult to draw definitive conclusions. For example, the comparative analysis of RNA-seq data from the macular and the peripheral retina in developing human eye has been analyzed previously by other groups. In addition, the involvement of the retinoic acid pathway is not a new or innovative finding, but rather supports published findings. Thus, it is crucial to contrast the data and results of the manuscript with a considerable amount of pre-existing published data, some of which was reported by the authors' laboratory. The main claim of the current manuscript appears to be the addition of snATAC-seq data. However, snATAC-seq alone, without integration of additional 'omics' data, does not greatly add to new mechanistic insights. While the authors occasionally allude to the consistency of their results with previous knowledge, they should explicitly articulate the new insights being reported in this manuscript by addition of new data.

Thank you for your insightful feedback and constructive suggestions on how to compare our findings with previously published data and to articulate new insights more effectively.

Our study's strengths lie in three primary areas:

First, the considerable increase in the number of cells profiled allows for an unprecedented resolution in exploring cellular developmental trajectories compared to prior studies. This enhancement in data volume facilitates a detailed examination of the complex processes underlying cellular development. Second, the simultaneous application of single-nuclei RNA-seq and ATAC-seq on the same nuclei, combined with advanced analytical tools, enables comprehensive integrative analyses. This approach aids in the identification of candidate gene regulatory networks and key transcription factors with improved sensitivity and accuracy, providing a deeper understanding of the developmental processes. Third, our care dissection and profiling of both the foveal and peripheral regions across an extensive range of developmental stages have led to the high-confidence identification of a wide array of genes and pathways distinguishing central from peripheral development.

The depth of our dataset has allowed us to identify a significant number of progenitor cells, overcoming previous challenges related to the scarcity of progenitor cell data. For example, our research profiles over 50,000 progenitor cells, revealing a smooth transition between early and later stages of progenitor cells, along with pathways specifically active during this transition phase. This discovery, not previously reported, enriches our understanding of cellular development. Moreover, the analysis of numerous neurogenic retinal progenitor cell (NRPC) populations has uncovered distinct progenitor cell subgroups for each retinal cell class. Previous studies only identified three clusters of transitional cells due to resolution limit (10.1016/j.celrep.2020.01.007). Further investigation of these progenitor cells across developmental stages has unveiled a novel hierarchical pattern of cell fate determination using ACs as an example. Moreover, we have discovered numerous transcription factors (TFs) specific to progenitor cells, including both well-characterized and novel TFs that merit further study.

The dataset also opens avenues for identifying genes differentially expressed between the foveal/macular and peripheral regions, potentially shedding light on the mechanisms of human foveal development. We observed notable differences between our findings in humans and those from other species, such as chickens, especially in the context of the RA and FGF8 pathways, highlighting species-specific developmental variations.

In response to the constructive feedback received, we have updated the abstract and revised several figures for enhanced clarity and impact. Recognizing the vast scope of our study and the unexplored aspects of our data, we emphasize that our findings lay a solid foundation for future research. We expect this rich dataset to serve as a valuable resource for both our team and the broader scientific community.

Specifically, to address reviewers' concerns, we have implemented many changes throughout the manuscript. For example, we reorganized content to improve logical flow and coherence, adjusting figure sequences to better illustrate our narrative. We now start with a discussion on global lineage in each figure, progressively narrowing down to specific details. Furthermore, we have included in our manuscript with comparisons to existing knowledge, incorporating previously published data on macula formation for

direct comparison with our findings. These enhancements, we believe, have significantly refined our manuscript, making it easier to follow by the readers.

Power: The authors utilize a relatively small number (12) of donors. The human donors (unlike laboratory models) have considerable variance to mask or exaggerate the differential outcome observed by the analysis being employed in the manuscript. The authors should either establish that the inter-donor variability is insignificant or must employ methods to mitigate such variations. Or, additional samples are needed.

Thank you sincerely for your invaluable suggestions. To tackle the power issue, we incorporated three additional samples sourced from two new donors. To show the inter-donor variability is insignificant, we plotted the UMAPs below. Three UMAPs, as depicted in the following figure, serve as evidence of the effectiveness of our data integration performance.

The first UMAP was colored by Lab ID, distinguishing between Rui Chen Lab (in blue) and Tom A Reh Lab (in orange). The second UMAP was colored by Sample ID, and the third UMAP was colored by sample age. This integration suggests that our data is not influenced by technical variations, individual differences, or batch effects. Instead, the

UMAP is primarily driven by cell types and ages, thereby further enhancing the reliability of our results.

When assessing the intricate dynamics of temporal and spatial gene expression during development, it is noteworthy that inter-donor variability is consistently insignificant in the context of human retina developmental samples. In the overall UMAP, we did not observe any cluster that is dominant by one or two samples. They are always a mixture of many samples. This observation is particularly compelling when considering the substantial fluctuations in gene expression both temporally and spatially throughout development.

Notably, the retina developments across various vertebrates are highly conserved. This finding serves as compelling evidence reinforcing the notion that such variability is indeed inconsequential in this specific developmental context. It is essential to recognize that inter-donor variability emerges as a pertinent concern predominantly in adult or organoids.

Rationale for using different analytical methods: The manuscript falls short in delivering a cohesive and comprehensive analysis of the data. The impression is given that the authors applied tools available in the literature without adequately justifying their choices or providing a detailed account of the rationale behind selecting specific tools over alternatives. A more thorough discussion is needed, addressing why certain tools were chosen, the significance of their application, and how the results from various tools contribute to a comprehensive and coherent narrative.

We appreciate your acknowledgment of our efforts to enhance the cohesion of our methodology. During the revision process, we worked to ensure uniform use of tools throughout our study by eliminating duplicate analyses. For instance, pseudotime analysis was omitted, as it duplicated the latent time inference derived from the velocity analysis.

In terms of tool selection, given the focus of our paper on retina development, we opted not to conduct an exhaustive benchmark of all available tools. Instead, we selected a subset of tools known for top performance based on benchmark publications. For data integration, we chose scvi-tools, supported by previous benchmarks (10.1038/s41592-021-01336-8). While we also experimented with Seurat for this purpose, our dataset proved too large for Seurat to handle efficiently. Due to space constraints, we omitted this detail from our manuscript.

It's essential to note the temporal aspect of tool application. In some instances, due to the limited number of tools that existed for multiome data at the time we prepared this manuscript, we employed tools available at the time of our analyses. For example, for gene regulatory analysis, we utilized Pando to infer the multiome Gene Regulatory Network (GRN). At that juncture, SCENIC+, another popular tool for GRN inference, had not yet been published. This timing consideration is crucial for a comprehensive understanding of our methodological approach.

RNA velocity analysis. The authors employed RNA velocity for trajectory inference analyses on multiple occasions. RNA velocity is based on measurements of spliced and unspliced mRNA transcripts to predict the direction of future gene expression changes in the cell. In scRNA-seq, RNA is collected from the whole cell, whereas in snRNA-seq, it is collected from the nucleus. This means that snRNA-seq may capture less cytoplasmic RNA and more unspliced nascent

RNA, additionally the RNA degradation rate therein may be replaced by the rate of RNA exporting the nucleus. There is concern about the applicability of this model to the snRNA-seq data. For example, in Figure 2B, the direction indicated by arrows in the upper left part of PRPC seems to be opposite to the real differentiation direction of PRPC in the Figure 2A. The authors need to make necessary adjustments to ensure the accuracy of the results. Did the authors perform multiple trajectory inference methods to analyze and validate their results? In addition, the authors have real chronological data of developing retina; could they indicate to what extent do their predicted directions of cellular differentiation correspond to the real direction of retinal development?

Thank you so much for your comments and we understood your concerns. Since we have 28 samples collected over a large time frame, we have enough time point to validate our velocity by correlation analysis between sample age and inferred latent time.

During revision, we add MGs and NRPCs when estimating velocity for PRPCs. This makes the velocity estimated pointing to the right directions.

In addition, our estimated velocity was estimated from both RNA-seq and ATAC-seq, which makes it more robust than traditionally RNA-seq alone. The left panel is showing velocities estimated with RNA-seq and the right panel is showing RNA-seq only. The left panel is much smoother and most of the velocities flow from cells collected from early samples to cells from late samples. The right one is messier.

In addition, during the revising process, we added statistical tests to show the latent time

Furthermore, to enable the ordering of cells from early to late stages, latent time was calculated based on PRPC velocities (Fig. 2C). As anticipated, latent time was significantly positively correlated with age (Pearson correlation 0.759, with an adjusted p-value of 0.000, see source data) (Fig. 2D). Additionally, as expected, using two-sample t-tests (see source data), cells in the macula consistently demonstrated a significantly larger average latent time compared to their peripheral counterparts from the same donor for all time points, except for PCW23, which had too few cells in the macula.

estimated from velocities are highly correlated with the sample age.

Submission of raw data. In the Data Availability part, the authors claimed that all raw data and processed data can be accessed in HCA Data Portal. But this reviewer can only see part of the processed data from that link rather than all the raw data. It only contains 205,619 cells after quality control instead of the original 296,158 nuclei they profiled. The authors are requested to submit the necessary raw data. In addition, the authors mentioned that they employed in house adult human retina derived data to perform the Major Class Annotation and Maturation Score Calculation. These are key factors that relate to the conclusions of the manuscript and therefore it is necessary for the authors to submit and clarify the raw data of adult cells they used.

Thank you sincerely for your valuable suggestions. We have implemented the necessary updates to the link, allowing users to seamlessly access and download raw data in the fastq format.

For individuals with limited programming experience, a user-friendly option is available at Cellxgene Data Exploration (<https://cellxgene.cziscience.com/collections/5900dda8->

2dc3-4770-b604-084eac1c2c82). This link provides a straightforward way to explore the data for both ATAC-seq and RNA-seq.

Alternatively, for more experienced programmers, the raw data in fastq format is accessible at Human Cell Atlas Data Exploration (<https://explore.data.humancellatlas.org/projects/581de139-461f-4875-b408-56453a9082c7>).

All in-house adult annotation data and adult histone modification data have been cited properly now and integrated into the Zenodo folder in the code availability section.

DATA AVAILABILITY

All processed data can be accessed through the HCA Data Portal at <https://cellxgene.cziscience.com/collections/5900dda8-2dc3-4770-b604-084eac1c2c82>. The raw data in fastq format can be accessed at <https://explore.data.humancellatlas.org/projects/581de139-461f-4875-b408-56453a9082c7>.

Adult data used for cell major class annotation, cell subclass annotation can be accessed at <https://zenodo.org/uploads/10866341>. Adult differentially accessible regions, adult histone modification regions, inferred gene regulatory network model, recovered dynamic models, identified differentially expressed genes model, and bigwig files for ATAC can be accessed at <https://zenodo.org/records/10866349>.

Here is a genome browser link that showcases the peak accessibility of the ATAC-seq data <https://genome.ucsc.edu/s/zhenzuo2/Single%20Cell%20Multi%20Domic%20Atlas%20of%20the%20Human%20Developing%20Retina>. Source data is provided.

Experimental validation: The study mentions many transcription factors and gene regulatory networks; however, most of these findings are inferred or previously predicted. Some experimental validation of new findings is essential for linking them to relevant biological context. The results of the analysis, particularly those included in the extensive lists in several supplementary tables, lack biological validation.

Thank you sincerely for bringing this to our attention. We value your insight and acknowledge the importance of testing all the transcription factors identified in this study. However, this task is extensive, with hundreds of factors requiring thorough examination. All TFs have distinct regulation effects, so only testing one or two of them will not solve the validation problem at all. To do justice to the depth and breadth of this investigation, it seems more appropriate to address the validation of each one of the transcription factors as a separate, independent study.

The validation process involves additional complexities, including the need for animal models such as mice or zebrafish. Regrettably, the inclusion of these models is beyond the current scope of our paper, which is primarily centered on the development of the human retina. We appreciate your understanding as we strive to maintain the focus and integrity of our research within the defined parameters of this study.

Limitations: The authors should provide a more balanced view by discussing more explicitly the limitations of their methodology and findings.

Thank you so much for bringing this to our attention. We are grateful for your feedback. In response to your observations, we have addressed the limitations directly within the discussion section for each identified point.

Additional specific comments:

1. To understand the analysis better, the authors should consider providing a workflow diagram. Thank you for bringing this to our attention and providing valuable suggestions. In order to enhance the clarity of the analysis for readers, we have incorporated an overall workflow diagram (refer to Fig. 1A).

Additionally, we have introduced diagrams for key analysis steps, including cell major class annotation, cell subclass annotation, and DEG analysis. These detailed diagrams are now presented in the supplementary figures section, offering a more comprehensive view of the analysis pipeline.

2. The manuscript mentions that a total of 51 cell types and 5 precursor groups were identified. How does this list compare with the comprehensive list of cell types curated by Sanes Lab and others in different organisms [Table 1 in

<https://www.annualreviews.org/doi/full/10.1146/annurev-vision-032621-075200>]. That table does not include cell types from developing retina and should be extended.

Thank you very much for bringing this matter to our attention. The cell type annotation process relied on the adult annotated reference available at <https://cellxgene.cziscience.com/collections/af893e86-8e9f-41f1-a474-ef05359b1fb7>. We selected this reference due to its remarkable sensitivity of 0.01%. It includes over 250K nuclei for single-nuclei, which is the same technology as our developmental data.

In this revised version, we aim to do a systematic comparison between cell types found in the development data with the comprehensive list of cell types. With the aim of enhancing readability, we have included an additional table in Supplementary Table 4 to summarize the number of cell types identified in our developmental data. Specifically, the table indicates how many cell types were found and how many were not found.

	A	B	C	D
1		Number of Cell Types in Adult	Number of Cell Types in both Adult and Development Data	Unque Cell Type in Adult
2	AC	33	31	AC 27 AC32
3	BC	13	12	DB4b
4	RGC	8	4	OPN4_RGC RGC8 RGC7 RGC6
5	Rod	1	1	
6	Cone	2	2	
7	HC	2	2	
8	MG	1	1	
9				
10				
11	Total	60	53	
12				

In summary, our analysis using the adult reference revealed a total of 60 cell types, with 53 of them successfully annotated within the developmental data. Notably, the cell types not found in the developmental data are primarily associated with ACs and RGCs. This absence may be attributed to the potential mixing of these cell types with others exhibiting similar gene expression patterns.

3. The manuscript also mentions that these cell types are hierarchical. It is imperative that the authors include a tree diagram of the cell types with chronological information as provided in Figure 4c or Figure 6g in Qui et al preprint from Shendure lab (<https://www.biorxiv.org/content/10.1101/2023.04.05.535726v1.full>)

Thank you for your comment. In response, we have included a tree diagram in Supplementary Figure 7. This illustration focuses on the AC lineage, demonstrating the heterogeneity within the progenitor level that ultimately leads to distinct subclasses among all ACs.

Initially, progenitors are depicted as the yellow cell cluster on the left side of the entire tree, serving as the root of the developmental tree. The presence of numerous overlapping yellow cells signifies the heterogeneity within the progenitors, influenced by spatial or temporal factors. Following activation and the involvement of certain TFs (not shown in the figure), progenitors begin to commit to the AC fate. However, heterogeneity persists within these AC progenitors, driven by subclass distinctions such as GABAergic AC progenitors, starburst AC progenitors, glycinergic AC progenitors, and dual AC progenitors.

Subsequently, these specific progenitors give rise to corresponding precursors, which eventually differentiate into mature AC subclasses. It's important to note that a similar

tree diagram can be generated for other major cell classes, such as BC, RGC, and so forth. However, for clarity and conciseness, we have chosen to focus on a single cell type in this illustration to convey the core concept and idea.

4. While annotating the cells into several major and sub cell types, it seems that the authors have utilized several tools and applied manual curation. However, the exact steps are unclear from the information provided in the main text and the supplement. To give an example, when the authors subdivided the Retinal Progenitor Cells (RPC) into Primary RPC (PRPC), Neuronic RPC (NRPC) and Mueller Gila (MG) how were the boundaries decided. I guess that some scaffolding was done by clustering those cells and the clusters were annotated with one of these sub cell types, but that crucial information is missing from the description.

Thank you very much for bringing this to our attention. In response, we have included a diagram to illustrate our annotation process for major classes, subclasses, and cell types, enhancing the clarity of the entire annotation procedure.

When we subdivided the Retinal Progenitor Cells (RPC) into Primary RPC (PRPC), Neuronic RPC (NRPC) and Mueller Gila (MG), the boundary was decided by clustering. After clustering, marker genes from previous publications were used to annotate each cell type. Then, the annotation was confirmed by cell cycle.

(A) Annotation Workflow Diagram (E)

5. The authors have used several pieces of information such as pseudo time, latent time and emission probability derived from tools such as multivelo, scvelo, monocle and cell rank. However, it is not clear what exact information from each of these tools were used. It is not clear if only one of the tools such as CellRank or scVelo would have been sufficient (e.g. CellRank

version 2 internally derives RNA velocity and similar information). Moreover, some of the tools have been mentioned in multiple sections in the materials and methods section making it more confusing why and in which order the tools were used. For example, MultiVelo has been mentioned in the subsection “Latent Time Inference”, “Multi-omic Dynamical Modeling” and “RNA Velocity Analysis and NRPC Fate Inference”. Why? My guess is that the tool was applied only once, and the results were reused multiple times. The authors should clarify that or simplify the methods sections by proper reorganization. Similar can be said about using monocle3 in “Maturation Score Calculation” and “Pseudotime Analysis”.

We appreciate that it could be confusing given the large number of tools was utilized in this study. We have reduced the method we used to make the analysis more transparent and clearer.

In summary, when inferring latent time, we use multivelo. When inferring cell fate, we use cell rank.

Here are the reasons behind this. We run “Multi-omic Dynamical Modeling” at the major class level, for example, on PRPCs, to infer latent time. However, it cannot be used on all cells because estimating velocities with both ATAC-seq and RNA-seq took too much memory and computing power. Currently, our 1TB memory can only take around 100,000 cells at a time, however, we have more than 200,000 cells. So, for cell fate inference analysis, which needed to be run on all cells, we only used RNA-seq and CellRank.

We removed monocle3 and all pseudotime analysis because we think they are duplicated with latent time estimated from velocity.

6. When the authors use a particular tool or does some analysis, the authors should make that analysis as much comprehensive as possible. For example, in the results section “AC Subclasses Formed from Specific Groups of NRPC at Different Times” the authors mention how a subgroup of Neurogenic RPCs differentiate to different Amacrine subtypes such as Starburst, GABAergic, Glycinergic and Dual ACs. But I guess that similar analysis can be done for NRPCs that differentiate to other major cell types such as photoreceptors, bipolar cells, and so on. Why was not similar analysis done?

I appreciate your insightful feedback. Allow me to elaborate on why we did not conduct similar analyses for all major classes:

1. Limited Cell Numbers: Some major classes inherently possess fewer cells in our dataset, making the analysis more challenging. For instance, Horizontal Cells (HCs) have a relatively low cell count compared to other types, with only around 500 HC0 and 900 HC1 cells. This limited sample size makes it difficult to infer velocity accurately based on such a small number of cells. A similar constraint is observed in Cone cells, with only around 300 S cones in our dataset.
2. Incomplete Developmental Trajectory: For certain major classes, we were unable to observe the entire cell fate specification process. For example, our dataset is not optimal for studying the developmental trajectory of Retinal Ganglion Cells (RGCs) since mature RGCs are already formed in our earliest time point. A similar limitation applies to Bipolar Cells (BCs) because BC differentiation occurs relatively late, and not all cells have completed differentiation.
3. Differentiation Time Window: The differentiation time window varies across major classes. We chose Amacrine Cells (ACs) as an illustrative example because they exhibit a relatively large specification window compared to other major classes.

Despite these challenges, we aimed to make our work as comprehensive as possible. During the revision process, we incorporated additional analyses, including Gene

Regulatory Network (GRN) analysis for each major class. Furthermore, we presented the estimated velocity for all major classes, ensuring a more inclusive exploration of our developmental model.

7. The authors should provide a smooth transition when changing from one result section to another.

Thank you sincerely for providing your valuable comments. In response to your feedback, we have enhanced the coherence of the result section by incorporating additional transition sentences. As an illustration, we have introduced sentences to facilitate a seamless transition from discussing PRPCs to NRPCs.

NRPC with Different Fates Show Distinct Gene Expression Patterns

After assessing heterogeneity within PRPCs, heterogeneity within NRPCs was examined. As the transition cell state, NRPCs give rise to all retinal neurons. Interestingly,

8. In all vertebrates, retinal cells are demonstrated to be produced in a conserved order with RGCs being differentiated first and Muller Glia at a late stage. Is it correct that, in Figure 2, the authors imply that MG are differentiated first? This needs a careful analysis and discussion. Again, validations are required for any such conclusions about trajectory of cell birth.

Thank you for your thoughtful comments. The consensus in the scientific community has long held that Müller Glia (MG) represents the last cell fate to be specified (references: [10.1523/JNEUROSCI.1624-07.2007](https://doi.org/10.1523/JNEUROSCI.1624-07.2007), [10.1016/j.preteyeres.2009.05.002](https://doi.org/10.1016/j.preteyeres.2009.05.002), [10.1038/ncb2115](https://doi.org/10.1038/ncb2115)). However, the precise timing of MG specification remains a topic of active investigation. A recent study claimed that MGs and all other cell types (such as BCs, and rods) can be detected in the presumptive fovea as early as 59 days through immunolabeling, yet it may take up to 180 days for MG to be uniformly detected in all retinospheres (DOI [10.1016/j.stemcr.2023.10.021](https://doi.org/10.1016/j.stemcr.2023.10.021)).

We are trying to imply that in the presumptive fovea, Müller glial cells (MGs) and all other cell types can appear extremely early. However, it takes much longer for other neighboring regions to also detect those cells. So, we did not imply that MGs are differentiated first. We are implying that the spatial differences are huge.

Minor comments:

1. The word “region” seems to have a dual meaning in this manuscript: 1) open chromatin region, and 2) region within the retina – macula or periphery. It would be good to use “spatial location” or something similar for the second case.

Major Downstream Analysis Workflow Diagram

Annotation Trajectory GRN Regional DEGs

Thank you for highlighting this concern. In response, we have adopted the term "region" to denote open chromatin regions and have employed "spatial location" to describe the specific spatial origin of the tissue, aligning with your suggested terminology. The figure above is an example of the changes we made.

2. The words reference/query and adult/fetal has been used interchangeably in the manuscript. It will be good to follow a consistent nomenclature; preferably adult/fetal.

Thank you for bringing this matter to our attention, and we wholeheartedly agree. In response to this concern, we have modified the terminology from "reference/query" to "Adult/Development" to ensure clarity for our readers.

3. How were gene activity scores in the UMAP of Fig. 1D generated?

Thank you for bringing this to our attention. In response, we have incorporated a dedicated section in the methods section, providing a detailed explanation of how the gene activity score was calculated for the benefit of our readers.

Here is the section we used to introduce gene score in the method section:

A gene score was calculated for each cell at the gene level to measure the degree of chromatin accessibility. This was achieved by summarizing peaks near the gene's transcription start site, gene body, as well as at the promoter and distal regulatory elements, and applying appropriate weighting. In this study, we calculated gene scores to quantify the openness of each gene using ATAC-seq data with ArchR cite{ArchR}. To estimate gene scores, we utilized ArchR with all default parameters, employing the `addGeneScoreMatrix()` function. During this process, gene scores were calculated through a weighted sum of accessibility within the gene body, as well as at the promoter and distal regulatory elements.

4. When referencing first time in line 94 on page 3, please cite the reference atlas.

Thank you for bringing this to our attention. We have included the following citation in the manuscript:

@article {Li2023.11.07.566105,

author = {Jin Li and Jun Wang and Ignacio L Ibarra and Xuesen Cheng and Malte D Luecken and Jiaxiong Lu and Aboozar Monavarfeshani and Wenjun Yan and Yiqiao Zheng and Zhen Zuo and Samantha Lynn Zayas Colborn and Berenice Sarahi Cortez and Leah A Owen and Nicholas M Tran and Karthik Shekhar and Joshua R Sanes and J Timothy Stout and Shiming Chen and Yumei Li and Margaret M DeAngelis and Fabian J Theis and Rui Chen},

title = {Integrated multi-omics single cell atlas of the human retina},

eolocation-id = {2023.11.07.566105},

```
year = {2023},
doi = {10.1101/2023.11.07.566105},
publisher = {Cold Spring Harbor Laboratory},
URL = {https://www.biorxiv.org/content/early/2023/11/08/2023.11.07.566105},
eprint = {https://www.biorxiv.org/content/early/2023/11/08/2023.11.07.566105.full.pdf},
journal = {bioRxiv}
}
```

5. Typo error: In Figure 2C, “Latet” should be “Latent”. Supplementary Fig.1F should be in 1E.

Thank you for bringing these typo errors to our attention. We have corrected the mentioned issues and conducted a thorough review of the entire manuscript to ensure it is free of any additional typos.

6. In Figures 2C and 2D, there is a lack of necessary introduction about the latent time. In other words, what does the value of the latent time mean? In addition, the conclusion for Figure 2D (Page 4 Line 122-125) lacks the necessary statistical description and is not rigorous enough.

To better elucidate the rationale underlying the concept of latent time and enhance readers' comprehension, an additional paragraph has been incorporated into the latent time calculation section of the method.

“The latent time serves as an intrinsic timer for each cell, providing an estimate of the cell's degree of differentiation during development or the elapsed time since differentiation initiation. Throughout development within a lineage, latent time was utilized to orderly categorize cells from early to late stages. In this study, latent time was inferred using MultiVelo ¹, a multi-omic single-cell velocity inference model.”

To provide more robust statistical support for our observations, two key considerations were addressed. Firstly, as anticipated, latent time exhibited a positive correlation with age, with samples from older donors displaying elevated latent time values. Additionally, an average higher latent time was observed in macula cells compared to their peripheral counterparts from the same donor.

To formalize these findings, a comprehensive statistical analysis was conducted.

“To verify the accuracy of the inferred latent time in representing the degree of development, we conducted a correlation analysis between PCWs (in numeric integer format) and latent time for all cells. Pearson's product-moment correlation coefficient was employed, with the null hypothesis positing that the correlation between PCWs and inferred latent time equals 0. The corresponding test results are available in the Source Data of Fig. 2D.

Subsequently, for each PCW, we assessed whether macula cells exhibit a higher latent time than peripheral cells for each week. The testing results can be found in the Source Data of Fig. 2D. To enhance statistical rigor, all p-values underwent Bonferroni correction to obtain adjusted q-values.”

7. Page 4 Line 135: The authors indicate Supplementary Fig.4A when they describe Module 2, but the Supplementary Fig.4A lacks necessary descriptions. It is confusing. I don't understand what this indicator is meant to illustrate.

Thank you for bringing this to our attention. Following the gene ontology analysis, we identified cell cycle-enriched terms among the gene ontology categories. To delve deeper, we specifically examined the cell cycle score within cells exhibiting a high module 2 score. Utilizing cell cycle-related genes, we estimated the cell cycle score, which represents the average expression values of these genes. The resulting cell cycle scores were then plotted on Fig. 4A to illustrate the overlap between cells with high cell cycle scores and module 2 scores.

However, in the revised version, we have removed this analysis. Upon further investigation and the inclusion of additional cells in these groups, the initially identified cell cycle gene-enriched module is no longer distinctly separated but is now part of module 1.

8. Supplementary Fig.4B-C: What does the value of the y-axis mean? In addition, it is necessary for the authors to describe in detail how they got the conclusions in Page 4 Line 141-143.

Thank you for bringing this to our attention. In the two panels in question, we aim to illustrate the process of cell fate inference for PRPCs. Initially, we performed clustering on all RPCs/MGs cells and subsequently conducted velocity analysis. To calculate the fate probability for each PRPC transitioning to either NRPCs or MGs, we designated the terminal states as NRPCs or MGs. The fate inference was implemented using CellRank, where we inferred the probability of each cell transitioning to a particular fate based on velocities and connectivities.

In Supplementary Fig. 4B, the clustering on the UMAP of RPCs/MGs is depicted. Velocity was estimated, and cell fate was inferred after manually determining the terminal state. We summarized the inferred fate probabilities for each cluster in a box plot, providing insight into which cluster has the highest probability of transitioning to NRPCs fate and which cluster exhibits the highest probability of transitioning to the MG fate (labeled as MG precursors). To enhance clarity, we have relocated this part to the main figure and added more detailed figure text explaining this process. Additionally, a diagram has been included to visually represent how the cell fate probability was estimated for each PRPC.

9. The description of some figure legends are too simple and lacks necessary explanations. For instance, lack of legend for colors: Figure 2I, Figure 3E, supplementary Fig.5C, Figure 4H.

10. Page 5 Line 174 : The authors showed identified differentially expressed genes of different cell cluster in Fig.3F and Supplementary Fig.5C, which could be interpreted as potential genes to drive neural type specification, however it does not seem to involve pathways. There is a suspicion of over-interpretation here.

Thank you for bringing this to our attention. To comprehensively address this concern, we have implemented several revisions. Firstly, we added a figure legend to all panels throughout the entire manuscript, extending beyond those mentioned above. Secondly, we standardized the colors used in the figure legend to ensure consistency across all panels. Thirdly, in order to enhance visibility and clarity, we increased the size of the figure legend, allowing readers to easily discern it alongside the other panels.

For Fig. 3F, subsequent to the identification of differentially expressed genes (DEGs) for distinct cell clusters, we conducted gene enrichment analysis to discern enriched pathways or biological processes. The results were incorporated into Fig. 3G. In this revised version, we have taken care to avoid over-interpretation and to provide robust evidence supporting our conclusions. For instance, the identified biological process

terms substantiate our fate predictions for progenitors, particularly for retinal ganglion cells (RGCs) and rods. Terms such as "postsynapse organization" were associated with neural retinal progenitor cells (NRPCs) destined for RGC fate, while terms related to "visual perception" were prominent in NRPCs committed to rod fate.

11. Figure 5D: Where did the histone modification data originate? Also please provide a description of the analysis process for this figure in the Methods section.

Thank you for bringing this to our attention. We have addressed the issue by adding the citation for the histone modification data and including the complete bed file in Supplementary Table 10. The histone modification information was sourced from the

publication referenced below.

@article{Wang_Cheng_Liang_Owen_Wang_DeAngelis_Li_Chen_2022, title={Single-cell multiomics of the human retina reveals hierarchical transcription factor collaboration in mediating cell type-specific effects of genetic variants on gene regulation}, volume={24}, DOI={10.1186/s13059-023-03111-8}, number={1}, journal={Genome Biology}, author={Wang, Jun and Cheng, Xuesen and Liang, Qingnan and Owen, Leah A. and Lu, Jiaxiong and Zheng, Yiqiao and Wang, Meng and Chen, Shiming and DeAngelis, Margaret M. and Li, Yumei and et al.}, year={2023}, month={Nov}}

In addition, we have incorporated a dedicated section in the methods to elucidate the processing steps undertaken to further refine and achieve our goals using this information.

12. Figure 5G: It looks confusing and not well described in the legend. The gray highlighted area indicates a cone-specific OCR region, yet this region doesn't seem to be a cis-regulatory module because it doesn't have a peak. Also, what do the asterisked peaks mean? The description for this figure is at Page 7 Line 263-268, but I'm not sure what the authors are trying to explain. The authors showed an example of OCRs near the OTX2 gene loci. Are they trying to show that these OCRs are associated with differential expression in Figure 5F? If so, they should provide the appropriate argumentation process.

Thank you for bringing this to our attention. In the revised version, we have enhanced the layout of this panel to facilitate easy information retrieval for readers. The primary message we aim to convey in this panel is to compare the linked peaks we identified

with existing knowledge. Ideally, our identified linked peaks should align with previously identified cis-regulatory modules from experiments.

To achieve this objective, we have chosen OTX2 as an example, given its extensive study in the field of neuroscience, leading to the identification of numerous cis-regulatory modules for this gene. The panel consists of four parts, arranged from top to bottom. The first part features a trace plot displaying the chromatin accessibility of the OTX2 nearby region, allowing for the comparison of chromatin accessibility across all major classes.

The second part delineates the locations of previously published cis-regulatory modules, with the names of these modules labeled at the top of each region. Further explanations for these modules can be found in the figure text section. The third section displays the linked peaks identified using our developmental retina data, and the last section presents the gene annotation for the same range.

In the first section, if a reported gene module overlaps with any identified linked peak, we have outlined it with a transparent black box to indicate consistency or discrepancy. Notably, 5 out of 13 previously reported cis-regulatory modules were identified as linked peaks in our data.

Several factors may contribute to inconsistencies with some modules. One reason could be that the mouse cis-regulatory modules were identified from cells outside the retina, potentially introducing variation due to different cell types. Another factor could be the species difference, as the cis-regulatory modules were identified from mice, while our linked peaks were identified from humans, potentially contributing to the differences. To avoid potential confusion regarding this panel, we have incorporated explanations in both the figure panel text and the methods section. These additional details are aimed at providing clarity to the reader, ensuring a better understanding of the intended message we seek to convey.

13. Page 8 Line 290 / Fig.6A and Supplementary Fig.9 and 10: The examines were performed in all cell types rather than only PRPC according to the figures. Page 9, line 329 and Figure 7A – AP3B1 seems to be high in all cell classes.

Thank you for bringing this to our attention, and you are absolutely correct. The analysis of Differentially Expressed Genes (DEGs) was conducted in a major-class-specific manner. However, for the presentation of identified DEGs, we aim to illustrate global gene expression differences. Consequently, the UMAP plot may include major classes that exhibit similar gene expression values.

Upon reflection, we acknowledged that the initial panels were confusing and did not effectively convey the intended message. One contributing factor was the use of violin plots, which, in some instances, failed to show gene expression differences due to sparsity. To improve this, we have replaced all violin plots with bar plots. Bar plots provide a clearer comparison of the percentage of cells with non-zero gene expression values.

Regarding AP3B1, its expression is consistently high across all major classes, with the highest expression observed in RPC/MG compared to other major classes."

14. Page 14, line 523 – enlist and cite cell cycle genes

Thank you for bringing this to our attention. To resolve this issue, we have streamlined our analysis pipeline. In the initial version, we reran the UMAP exclusively on RPCs/MGs cells, excluding cell cycle genes, to enhance resolution. However, in the revised version, to avoid potential confusion arising from repeated UMAP reruns, we chose to use uniform global UMAP. This adjustment was made to improve reader comprehension without sacrificing essential details. Consequently, in the revised version, we have removed the step of removing cell cycle genes. So, this issue is well addressed now.

15. Page 14, line 525 – Six PCW should be five PCW

Thank you for your insightful questions. You are correct, and we have implemented the corresponding change.

16. Page 14, line 556 – cite “established marker genes”

Thank you for bringing this to our attention. We have now included the following citation in our manuscript:

@article{thomas_timms_giles_harkins_perry_lyu_hoang_qian_jackson_bahlo_blackshaw_et,

title = {Cell-specific cis-regulatory elements and mechanisms of non-coding genetic disease in human retina and retinal organoids},

author = {Thomas, Eric D. and Timms, Andrew E. and Giles, Sarah and Harkins-Perry, Sarah and Lyu, Pin and Hoang, Thanh and Qian, Jiang and Jackson, Victoria E. and Bahlo, Melanie and Blackshaw, Seth and et al.},

year = 2022,

journal = {Developmental Cell},

volume = 57,

number = 6,

doi = {10.1016/j.devcel.2022.02.018}

}

17. Page 15, line 565 – percentile better than quantile

Thank you for your insightful questions. You are correct, and we have implemented the corresponding change.

18. Page 15, line 575 – I do not understand why we need to fit, as mean and standard deviation completely determines the density. Right?

Thank you for your insightful questions. You are correct; we can directly plot the density without engaging in any modeling or fitting processes. However, we opted to fit the data with a normal distribution, aligning with established knowledge as cited (10.1016/S0166-2236(00)02028-2). Fitting the cell birth rate with a normal distribution is a standard practice in mouse retina development, providing a useful framework for analysis.

Additionally, employing the normal distribution allows for a smoother representation of

the observed cell birth rate, facilitating straightforward comparisons of cell birth order and proportions for the reader. It is more like a de-noising and smoothing approach to use.

19. Page 15, line 592-595 – It would be good to show monocle pseudotime in a plot somewhere.

Thank you for pointing that out. Upon review, we couldn't locate the specified section on Page 15, lines 592-595, either in the main text or supplementary table. In the revised version, we have completely removed the Monocle pseudotime segment. We think pseudotime estimated from Monocle3 is duplicated with latent time estimated from velocities. Instead, throughout the entire manuscript, we now utilize estimated latent time to order cells within each lineage. These changes have been made to enhance the clarity and consistency of the analysis pipeline.

20. Page 18, line 679 – “g Profiler” -> “g:Profiler”

Thank you for bringing it to our attention. Your observation is entirely accurate. We have implemented the necessary changes accordingly.

21. Page 20, line 766 – Fig. 1H should be Fig. 1C

Thank you for bringing it to our attention. Your observation is entirely accurate. We have implemented the necessary changes accordingly.

22. Page 21, lines 792-793 – “UMAPs on the right represent cells with highest module scores for each of the identified modules” -> “UMAPs on the right show a cell with the color of the module among all identified modules that has highest score for that cell” or something like that. Also, these UMAPs are confusing. I would suggest including all cells but showing irrelevant cells in gray color. BTW, the methods section should explicitly mention how these module scores are computed.

Thank you sincerely for your valuable suggestions. We have modified the wording as per your recommendations and made corrections based on the paper's method section that we cited. Instead of plotting a cell with the color of the module among all identified modules that have the highest score for that cell, we now plot module scores for all cells. Additionally, we have incorporated a dedicated section in the Methods section to elucidate the calculation process of the module score, as follows:

In our case, we identified three modules. For example, for each cell, the module 1 score is a , the module 2 score is b , and the module 3 score is c . Then, $a + b + c = 0$. If a cell has a high expression of genes in module 1, it will have a large 'a' value, and the 'b' and 'c' values will be small. In the UMAPs, we plot the module 1 score for all cells, the module 2 score for all cells, and the module 3 score for all cells. This allows the reader to observe all these values from these three UMAPs.

Gene Module Score

Additionally, the gene module score was calculated. The gene module score is a per-cell summary score that measures the general pattern of expression for genes in a module. It is the cell-loadings of a single component PCA using smoothed count matrix of all the genes within each module.

23. Page 22, line 866 – how are motif deviations computed?

Thank you for your inquiries. The motif deviations referred to in our study signify the usage patterns of motifs. Specifically, a deviation is a bias-corrected metric that quantifies the extent to which the per-cell accessibility of a particular feature (in this case, a motif) diverges from the anticipated accessibility, calculated based on the average accessibility across all cells or samples. The computation of motif deviations was carried out using the `addDeviationsMatrix()` function within the ArchR package in R. A comprehensive account of this procedure is available in the Methods section.

ATAC Motif Enrichment and Motif Deviation Analysis

Motif enrichment and deviation analyses were conducted on the pseudobulk peak set. To annotate the peaks, we utilized the Catalog of Inferred Sequence Binding Preferences (CIS-BP) motif from ChromVAR[68]. Furthermore, we computed the chromVAR deviation scores for these motifs using the implementation in ArchR. Subsequently, the motifs were ranked based on the $-\log_{10}(\text{P-adjusted})$ Motif Enrichment. To identify genes with highly correlated gene expression and motif deviations, 'correlateTrajectories()' in ArchR was used with default parameters.

To make the “method” section met word limit, we moved some of the method details into “SUPPLEMENTARY NOTE” section.

24. Page 23, line 884 – Log2 of fold change?

Thank you for bringing this to our attention. The term 'Log2' should represent log base 2-transformed fold changes. This has been rectified in the revised version.

25. Page 23, line 889 – What does “red” group of DARs signify?

Thank you for your inquiry. In this section, our objective is to assess whether the linked peaks we have identified exhibit significantly higher levels of H3K4me2 and H3K27ac. To accomplish this, we initiated peak calling on H3K4me2 and H3K27ac data derived from the adult retina. This process was executed by the first author, as detailed in the paper with DOI 10.1186/s13059-023-03111-8, where the peak calling methodology is thoroughly explained. Subsequently, we determined the percentage of overlap between these histone-modified peaks and our identified developmental linked peaks, as indicated in the red column. To investigate the hypothesis that the linked peak regions exhibit significantly higher signals than the background, we randomly sampled an equivalent number of linked peaks from the entire set and compared the overlapping proportions. A detailed description of this procedure can be found in the Methods section. Additionally, we have refined the wording in this panel to enhance reader comprehension of the rationale behind our approach.

26. Page 23, line 892 – How do you obtain functions of cis-regulatory regions? I guess it is different than GO terms mentioned in methods section.

To elucidate the functionality of cis-regulatory regions, we employed the Gene Ontology Enrichment Analysis (GREAT) tool.

@article{McLean_Bristor_Hiller_Clarke_Schaar_Lowe_Wenger_Bejerano_2010, title={Great improves functional interpretation of cis-regulatory regions}, volume={28}, DOI={10.1038/nbt.1630}, number={5}, journal={Nature Biotechnology}, author={McLean, Cory Y and Bristor, Dave and Hiller, Michael and Clarke, Shoa L and Schaar, Bruce T and Lowe, Craig B and Wenger, Aaron M and Bejerano, Gill}, year={2010}, month={May}, pages={495–501}}

@article{Tanigawa_Dyer_Bejerano_2022, title={WHICH TF is functionally important in your open chromatin data?}, volume={18}, DOI={10.1371/journal.pcbi.1010378}, number={8}, journal={PLOS Computational Biology}, author={Tanigawa, Yosuke and Dyer, Ethan S. and Bejerano, Gill}, year={2022}, month={Aug}}

To enhance the scholarly integrity of our manuscript, we have incorporated the relevant citations during the revision process. In the meanwhile, we have introduced a dedicated section within the Methods to comprehensively address this aspect.

Peak Annotation

Peaks were inferred with GREAT [78, 79] to predict their biological functions. GREAT was used by providing the peaks as a bed file and the whole genome as background region through <https://great.stanford.edu/great/public/html/>.

To make the “method” section met word limit, we moved some of the method details into “SUPPLEMENTARY NOTE” section.

27. Page 23, line 898 – Please cite from where you obtained histone modification information. We obtained the histone modification information from the publication mentioned below.
- @article{Wang_Cheng_Liang_Owen_Wang_DeAngelis_Li_Chen_2022, title={Single-cell multiomics of the human retina reveals hierarchical transcription factor collaboration in mediating cell type-specific effects of genetic variants on gene regulation}, volume={24}, DOI={10.1186/s13059-023-03111-8}, number={1}, journal={Genome Biology}, author={Wang, Jun and Cheng, Xuesen and Liang, Qingnan and Owen, Leah A. and Lu, Jiaxiong and Zheng, Yiqiao and Wang, Meng and Chen, Shiming and DeAngelis, Margaret M. and Li, Yumei and et al.}, year={2023}, month={Nov}}

It's important to note that the data was not directly downloaded from the supplementary section of the article. Instead, we acquired the information by reaching out to the first author of the paper, who graciously shared it with us. This additional data is now included in the supplementary Table 10 for reference.

28. Page 23, line 900 – Provide more information about linked peak – how are they computed? Thank you for bringing this to our attention. During our revision, we have incorporated additional details regarding the calculation of linked peaks in the method section. To briefly summarize, linked peaks were computed by assessing the correlation between gene expression and chromatin openness. This calculation was executed using ArchR

with default parameters, and specifically, the function ArchR::addPeak2GeneLinks() was employed.

29. Page 23, line 898 – Please cite from where you obtained cis-regulatory information.

The following citation was added to address where we find this information:

@article{Chan_Lonfat_Zhao_Davis_Li_Wu_Lin_Ji_Cepko_Wang_2020, title={Cell type-

Identification of Peak-to-Gene Linkage

The identification of peak-to-gene linkages entails examining pairs of peaks and genes that display a robust correlation between peak accessibility and gene expression. In this analysis, we established a maximum permissible distance of 250,000 base pairs for each gene while searching for correlated peaks. Additionally, a minimum peak-to-gene Pearson correlation coefficient of 0.45 was set. These criteria were implemented using the "addPeak2GeneLinks()" function in ArchR, and the results were subsequently extracted using the "getPeak2GeneLinks()" functions in the same tool.

and stage-specific expression of OTX2 is regulated by multiple transcription factors and cis-regulatory modules in the retina}, DOI={10.1242/dev.187922}, journal={Development}, author={Chan, Candace S. and Lonfat, Nicolas and Zhao, Rong and Davis, Alexander E. and Li, Liang and Wu, Man-Ru and Lin, Cheng-Hui and Ji, Zhe and Cepko, Constance L. and Wang, Sui}, year={2020}, month={Jan}}

In addition, to make it clear, we added the information we used from this publication into our source data section so the reader can easily access it.

To make the "method" section met word limit, we moved some of the method details into "SUPPLEMENTARY NOTE" section.

30. Page 23, line 918 – Explain "priming in total gene time"

Thank you so much for pointing that out. In MultiVelo, time measurements have two distinct categories: global latent time and gene time. Global latent time serves as a quantification corresponding to the predicted developmental progress, whereas gene time is computed for each gene using ordinary differential equations. Specifically, these equations are modeled to estimate the time required for four possible gene states: primed, coupled on, decoupled, and coupled off. These states are defined by the intricate interplay between gene expression and chromatin openness patterns.

Gene time represents the cumulative duration of these four potential stages. Within this framework, priming in total gene time refers to the percentage of the 'primed' stage's proportion in the overall gene time. To delve into this aspect further, we have incorporated an additional section in the methodology to provide a comprehensive discussion, ensuring clarity on this matter.

The latent time acts as a cell's intrinsic timer, estimating the cell's degree of differentiation during development or how much time has passed since differentiation began. It aggregates the time assignments per gene (gene time), calculated within scVelo's dynamical model [63], onto a global scale that accurately approximates the internal clock of individual cells. During development, within a lineage, we used latent time to

To make the "method" section met word limit, we moved some of the method details into "SUPPLEMENTARY NOTE" section.

31. Fig 1c/d – Consider using better color scheme

Thank you so much for this insightful idea. We have made updates to ensure consistency in the color scheme across all gene expression and gene score figures. Additionally, for improved data visualization, we have transitioned to imputation gene scores to enhance the signal and create better contrast across the figures.

32. Fig 1H – Consider using continuous time scale instead of discrete. Preferably using number of days past conception.

Thank you so much for pointing that out. We completely agree, and we have implemented all the necessary changes to align with this idea. Specifically, the x-axis for all panels has been adjusted to represent the number of days post conception, and the y-axis has been standardized so that all sub-panels share the same range. In addition, for this panel, we have updated the cell maturation score estimation process to enhance method clarity and improve interpretability.

33. Fig 2E – Why is the blue annotation bar on the left of heatmap not extending all the way to the bottom? Are there genes outside the three modules?

Thank you so much for pointing that out. Indeed, that is correct. Using a correlation threshold, only genes that are dynamic with latent time will be used to find gene modules. In total, 2,860 latent-time-correlated genes were identified. When finding modules, one argument we used is 'core_only=True.' This argument filters out genes that are not highly correlated with any identified modules, so that finally, each identified gene module will only include genes that have a high correlation with each other. In our

data, 2,055 out of 2,860 genes were assigned to three modules. So, in summary, genes that do not exhibit similar patterns with any modules were intentionally left unassigned,

resulting in a scenario where some genes do not belong to any modules. As a result, the annotation bar will not extend to include all genes.

34. Fig 2G – provide gene expression color legend

Thank you for bringing this to our attention. To depict the temporal changes sorted by the estimated latent time, the gene expression values have been z-score transformed for each gene across all cells. Additionally, in response to your feedback, a gene expression z-score color legend has been incorporated into the figure during the revision. You can see the color legend was added to the right bottom of the panel.

35. Fig 2H – why is not the GRN showing edges between genes?

Thank you for your suggestion; we appreciate the input. We have implemented the idea and generated an example of the NRPC GRNs. In the displayed figure, we acknowledge that the edges are somewhat overcrowded, making it challenging to discern individual connections. To maintain clarity and organization, we have decided not to plot edges between genes directly. Additionally, to ensure accessibility for all readers, we have included a supplementary table summarizing all the regulations. This allows readers to easily search for specific regulations of interest. We also upload the Seurat objects used for this GRN so the user can plot those figures by themselves (<https://doi.org/10.5281/zenodo.10806576>).

36. Fig 3E – provide gene expression /emission probability color legends
 Thank you for identifying this issue. Now, the gene expression and fate probability legends have been added.

37. Fig 3G – why is not the GRN showing edges between genes?
 Thank you for your suggestion; we appreciate the input. We have implemented the idea and generated an example of the NRPC GRNs. In the displayed figure, we acknowledge that the edges are somewhat overcrowded, making it challenging to discern individual connections. To maintain clarity and organization, we have decided not to plot edges between genes directly. Additionally, to ensure accessibility for all readers, we have included a supplementary table summarizing all the regulations. This allows readers to easily search for specific regulations of interest. We also upload the Seurat objects used

for this GRN so the user can plot those figures by themselves (<https://doi.org/10.5281/zenodo.10806576>).

38. Fig 3J – Why do the UMAPs look different than 3A?

Yes, we have adjusted the UMAP with a specific focus on BC progenitors to reveal more consistent patterns. Subsequently, one panel has been omitted due to challenges encountered during the velocity analysis rerun with both clusters of BC progenitors. Specifically, one cluster exhibited a less smooth velocity flow into BC precursors. To address this, we opted to remove one of the BC progenitor clusters and only annotated

cells with a high fate probability, ensuring a more accurate representation of the analysis results.

39. Fig 4B – Can the two plots have same Y-axis scale for better comparison of the relative value of density?

Thank you for highlighting this concern, and we fully agree. We recognize that inconsistent Y-axes between figures may lead to misleading conclusions. In response, the revised version standardizes the Y-axis across the two sub-panels. Furthermore, we have replaced the previous line chart with a stacked bar plot to facilitate a more accurate comparison of cell type birth orders. This modification ensures that the dynamic nature of cell types is better illustrated in the figures.

40. Fig 4D – Are the UMAP coordinates of these two plots different from panel A?

Yes, we have reconfigured the UMAP specifically focusing on AC progenitors to unveil more consistent patterns. The remade UMAP resulted in a much clearer cell ordering compared to the one showing in all ACs. In the revised manuscript, for this particular section, we opted to utilize the same UMAP (the same UMAP used for all ACs), as the patterns are clearly discernible without the need for a remake.

During revision, we tried to use the same UMAP as much as we can on subset of cells. These adjustments were made to enhance reader comprehension without compromising resolution significantly.

As shown in the figure above, now panel (A) and panel (F) showed the same UMAP. 41. Fig 4H – Are all the points in a single UMAP or the three modules have separate UMAPs? Thank you for bringing this to our attention. The Gene Regulatory Networks (GRNs) presented in the manuscript are all visualized within a single UMAP generated based on regulatory effects. To enhance clarity and avoid overwhelming the reader with the large number of regulations, we have partitioned the entire GRN into distinct modules. In the revised version, the layout of the UMAP has been modified to encircle different modules with dashed lines. This modification ensures that the UMAP is presented in a clear and unambiguous manner.

42. Fig 5 – It is well known that as the cells differentiate, the number of open chromatin peaks are reduced. There should be plot to show the same in this data.

Thank you for your valuable suggestions. We are enthusiastic about the idea. Our attempt to compare the number of peaks identified at each time point did not reveal consistent patterns. Several factors may contribute to this observation. Firstly, during differentiation, lineage-specific factors play a dual role by shutting down previously accessible loci while simultaneously opening new chromatin loci to establish a lineage-specific gene regulatory network (10.15252/embr.202051644). Secondly, while ATAC-seq enables the direct measurement of open and closed chromatin, comparing the global degree of chromatin openness between samples can be challenging and biased. Thirdly, there are confounding factors such as the number of cells sequenced, sequencing depth, and potential biases introduced by the peak calling tools utilized, which may influence the conclusions drawn from the analysis.

43. Fig 6A – which sub panel is for macula, and which is for periphery?

The information regarding 'macula' and 'periphery' was initially incorporated into the title of each sub-panel. However, due to a small font size, clarity for readers might have been compromised. In the revised sub-panel, we have increased the font size of the title to ensure clear visibility, allowing readers to easily discern whether the UMAP corresponds to macula cells or periphery cells.

Here is a screenshot of the current figure layout.

44. Supp Fig 1A – the figure legend and text have reversed meaning of the colors.

Thank you for bringing this to our attention. We have eliminated the 'reference' and 'query' legends previously employed to reduce potential confusion. Instead of using 'reference' and 'query,' we now utilize 'development' and 'adult' to illustrate the cell annotation process, whereby the developmental data is annotated with adult data.

Step 3: Development and Adult Data Integration

45. Supp Fig 1C/D – the UMAP does not look like the bown part of UMAP in SF1B

Thank you for bringing this to our attention. In the manuscript, you may notice that the UMAPs did not match with each other. This discrepancy arises from rerunning UMAP on a subset of cells, causing the global UMAP and the UMAP of the subset to be inconsistent. The positive aspect is that rerunning UMAP on the subset of cells allows for capturing more details with higher resolution compared to the global UMAP. However, the drawback is that readers may be confused by the mismatched UMAPs.

For instance, in Supp Fig 1C/D, the disparity between the UMAPs is attributed to the fact that the UMAP in SF1B was generated using both adult and developmental cells, while SF1C/D focused solely on developmental RPCs/MGs. To achieve a finer resolution for a specific cell subset, we conducted a new UMAP analysis, leading to the observed inconsistency. In the revised version, we have enhanced the figure layout and included a diagram to clarify the re-running of the UMAP step.

46. Supp Fig 1F – F missing on figure.

Thank you for bringing this to our attention. The figure and figure text were not matched. In Supp Fig 1E, there were two sub-panels. One was the gene expression heatmap, while the other one was the gene score heatmap. In the revised version, we have

reorganized Supplementary Figure 1E into two separate panels, allowing readers to examine each independently.

47. Ensure color legends in all figures.

Thank you for bringing this to our attention. In the revised figure, we have added color legend for all figures. Here are examples of the color legend we added. For example, we add color legends to all UMAPs to show color scales.

In addition, we have ensured consistency by aligning the color legends across all figures. For example, for spatial locations, we now use red to represent the macula, blue for the peripheral region, and yellow for the entire eye. Additionally, all colors used for

each major class in the manuscript are now standardized across all figures for better coherence.

Reviewer #4 (Remarks to the Author):

I co-reviewed this manuscript with one of the reviewers who provided the listed reports as part of the Nature Communications initiative to facilitate training in peer review and appropriate recognition for co-reviewers.

Reviewer #5 (Remarks to the Author):

I co-reviewed this manuscript with one of the reviewers who provided the listed reports as part of the Nature Communications initiative to facilitate training in peer review and appropriate recognition for co-reviewers.

REVIEWERS' COMMENTS

Reviewer #2 (Remarks to the Author):

In this revised manuscript, the authors have provided additional experimental data, analysis and discussion to address all my concerns. Congratulations on this elegant study and I expect the dataset would be of high interest to many researchers in the field.

Reviewer #3 (Remarks to the Author):

The authors have done an excellent job in responding to the reviewers' comments. Overall, this is an excellent resource article for scientists in the field. Two minor points:

Several of the conclusions on gene regulatory networks are based only on single cell expression and ATAC-seq data. Single cell data is qualitative, and not quantitative. The authors should include that as one of the limitations. Hopefully, in future manuscripts, the authors will integrate epigenomic marks and Hi-C data, which can provide more fine details of networks.

This reviewer still does not agree with the use of term "multi". The authors have generated the "atlas" by using two datasets. Others have used this "term" does not mean it is right. Multi is used for "more than two". A quick Google search of "Multi" comes up with the following in 4 different dictionaries:

Multi- Definition & Meaning

Merriam-Webster

<https://www.merriam-webster.com/dictionary/multi->

The meaning of MULTI- is many : multiple : much. How to use multi- in a sentence.

MULTI Definition & Meaning

Dictionary.com

<https://www.dictionary.com> › browse › multi-

4 days ago — a combining form meaning “many,” “much,” “multiple,” “many times,” “more than one,” “more than two,” “composed of many like parts,” “in many ...

Multi- Definition & Meaning | Britannica Dictionary

Britannica

<https://www.britannica.com> › dictionary › multi-

MULTI- meaning: 1 : many much; 2 : more than two.

Definition of 'multi-' from Collins English Dictionary

multi-

(mʌlti-)

prefix

Multi- is used to form adjectives indicating that something consists of many things of a particular kind.

...the introduction of multi-party democracy.

...a multi-million-dollar outfit.

Reviewer #4 (Remarks to the Author):

Reviewer #5 (Remarks to the Author):
